# Integration of Freeplay-Induced Limit Cycles Based on a State Space Iterating Scheme

**Xiangyu Wang, Zhigang Wu * and Chao Yang**

School of Aeronautic Science and Engineering, Beihang University, Beijing 100191, China; wangxyflying@buaa.edu.cn (X.W.); yangchao@buaa.edu.cn (C.Y.)

*   Correspondence: wuzhigang@buaa.edu.cn; Tel.: +86-010-82317510

**Featured Application: To derive all possible limit cycles of a freeplay nonlinear dynamic system using time integrations with massive combinations of initial conditions efficiently and to analyze the limit cycle oscillation (LCO) stability of the initial conditions.**

**Abstract:** Time integration is commonly used to obtain accurate system responses, such as the limit cycle oscillations (LCOs) for an aeroelastic system with freeplay. However, the integrations that start with various initial conditions (I.C.s) are usually studied case by case, so only a few system states can possibly be focused on. This paper proposes a state space iterating (SSI) scheme to find LCO solutions using time integration by using another method. First, a large number of arbitrary I.C. cases are used for time integrations, but only a very short integration time is required for each I.C. case. Second, system behaviors are depicted visually through a method that combines a modified Poincaré map and Lorenz map, in which the LCO solutions are found as fixed points via visual inspections. To verify the SSI scheme's ability to find LCOs, a typical plunge–pitch wing section is established numerically. Time integrations with both the classic scheme and the proposed SSI scheme are carried out. The LCO results of the SSI scheme are well-aligned with those from the classic scheme. The SSI scheme visualizes the patterns of system responses using arbitrary I.C. cases and analyzes the LCO stability, which provides more mathematical insights into an aeroelastic system with freeplay.

**Keywords:** time integration; initial condition; limit cycle oscillation; freeplay; Poincaré map; Lorenz map

## 1. Introduction

Nonlinearities inevitably occur in most real dynamic systems and can induce abundant nonlinear behaviors such as limit cycle oscillations (LCOs), quasi-periodic motions, and chaos. Aeroelasticity is a discipline that usually studies aircraft dynamic systems that involve elastic structures subject to aerodynamic forces, inertia forces, and/or control systems [1]. Over the decades, scholars have been aware that when freeplay—one of the most ubiquitous and typical nonlinearities on the connecting parts of an aeroelastic system—occurs, persistent and complex nonlinear behaviors will arise and affect the system's safety [2,3]. Therefore, the development of effective and efficient methods to analyze the nonlinear behaviors in an aeroelastic system with freeplay has become a major subject of concern.

Analyses of nonlinear behaviors involve at least two important aspects: (1) obtaining the nonlinear responses in the form of time histories or some parameters that can re-construct the responses (e.g., the amplitudes and frequencies for an LCO) to understand "what is happening for the system", and (2) obtaining as many and as detailed physical and mathematical insights as possible to suggest "why it happens or how important it

is"—for example, analyzing the stability of an LCO or determining the type of an LCO bifurcation.

Various methodologies are available for obtaining nonlinear responses, such as the harmonic balance (HB) method, continuation, and time integration, among which time integration is recognized as the benchmark for other methods due to its high accuracy and compatibility with all kinds of nonlinearity [1,4]. Time integration solves equations of motion under a set of discrete time instances using various finite difference schemes such as Newmark, Euler, and Runge–Kutta, thus providing a straightforward approach to obtain accurate solutions. The accumulating errors due to finite difference have been addressed by many researchers via schemes using adaptive time steps, such as the fourth–fifth-order Runge–Kutta (RK45) scheme and the precise integration method (PIM) [4–6]. In addition, the difficulty when dealing with non-smooth nonlinearities such as freeplay can be handled by a combined scheme of the Hénon method and RK45 (Hénon–RK45) [4,7]. The kernel of Hénon–RK45 is to find all "switching points" where the non-smooth effects arise and move the system states to those switching points accurately. Therefore, time integration provides excellent access for the nonlinear responses of aeroelastic systems with freeplay.

However, several weaknesses of time integration are still seen in the following aspects: (1) Time integration falls short when analyzing the effects of initial conditions (I.C.s), which are quite significant for understanding nonlinear behaviors. The most widely adopted process is to assign only one principal system state a series of non-zero values and study the LCO behaviors case by case. For example, Padmanabhan and Dowell [8] studied an all-moving surface model and a wing-store model with various structural nonlinearities, such as cubic stiffness, cubic damping, and freeplay. In their study, less than one hundred I.C.s were tested in time integrations for the second model, while that number for the first model was only one. In each I.C. case, only the pitch angle was specified as a non-zero value, and the integrations with different I.C.s were tested case by case, which was quite time-consuming and only partially confirmatory. However, they did determine an important conclusion: some aeroelastic models can be very sensitive to the I.C., which might lead to completely different system behaviors such as decaying to zero, entering into an LCO, or achieving oscillatory divergence. Tang and Dowell [9] studied three cases of I.C.s in Section VI.A.3 in their paper. In all three cases, only one state—the displacement at the wing-tip for the first case and at the store pitch angle for the second and the third cases—was specified as a non-zero value, and the authors found that the LCO behaviours could be very sensitive to both the I.C.s and flow velocity. Tang and Dowell [10] also found that the flutter instability boundary was significantly dependent on the initial pitch angle. However, the authors only assessed a few non-zero initial pitch angles. Many other studies of the I.C. effects on LCO behaviors followed a similar method to that mentioned above [11,12].

(2) Time integration inevitably loses efficiency to some extent when dealing with freeplay due to the need to frequently search and verify the switching points of the system. Moreover, aperiodic/quasi-periodic motions and chaos are commonly seen in aeroelastic systems, even for simple wing section models [13,14], and a lengthy transient phase may also appear before a steady LCO. As a result, a wide range of time intervals could be required for confirming an LCO in time integration [4]. This presents a dilemma, where accurate LCO results require all switching points to be located along with a long integration time, while a large number of I.C. cases and airspeed cases demand high efficiency. To balance the accuracy and efficiency is, therefore, not an easy task.

This dilemma spurred us to seek another way of using time integration when dealing with aeroelastic systems with freeplay. First, the feature of high accuracy in time integration should remain since this feature is the basis upon which time integration serves as the benchmark for other methods. Therefore, calculations were carried out using the Hénon–RK45 method in this paper. Second, inspired by the application of the Poincaré map ([15] and Section 3.4 in [16]) on aeroelastic systems with freeplay [10,17–20], as well as the

heuristic and visualized method, using a Lorenz map (Section 9.4 in [21]), we propose a novel scheme to implement a modified Poincaré map using a Lorenz map. In this way, we can visualize the spatial patterns of all system states through massive I.C. cases and use a short integration time for each I.C. case. Moreover, fixed points of the modified Poincaré map that represent the LCO solutions can be easily obtained through a visual inspection of the graph of state patterns, which is very efficient. The LCO stability could also be confirmed in the same way.

Because this new scheme requires an iteration procedure in state space, we named it the state space iterating (SSI) scheme and applied it to a typical plunge–pitch wing section subject to unsteady aerodynamic loads. The proposed SSI scheme yields two LCO branches for the wing section, which are well aligned with those derived from the classic time integration using the Hénon–RK45 method. The highlights of the SSI scheme are as follows: (1) The spatial patterns of system states are clearly pictured, so one can easily find LCO solutions and confirm the stabilities directly and visually; (2) the integration time for each I.C. case is reduced to a few LCO periods, so a large number of I.C. cases would not decrease the calculation efficiency; and (3) the proposed method can be easily extended to all other kinds of structural nonlinearities and applied to an aeroelastic system with higher dimensions.

This paper is organized as follows: Section 2 presents the state space equations for a general aeroelastic system with freeplay and introduces the classic Hénon–RK45 method. Section 3 introduces the proposed SSI method and explains how the Poincaré map and Lorenz map are modified, applied, and finally form the SSI scheme together. A plunge–pitch aeroelastic model is established in Section 4. Then, an LCO analysis based on the Hénon–RK45 method is implemented in Section 5. Time integrations based on the SSI scheme are carried out in Section 6, along with the discussions and a comparison to the Hénon–RK45 method.

## 2. State Space Equations and the Hénon–RK45 Method

### 2.1. State Space Equations of an Aeroelastic System with Freeplay

The equations of the motion of an aeroelastic system with freeplay can generally be expressed in the form of a set of piecewise linear state space equations (e.g., Equation (5) in [3], Equation (8) in [4], and Equation (1) in [13]):

$$\dot{\mathbf{x}} = \mathbf{A}_0 \mathbf{x} + \mathbf{b} f_{\mathrm{FP}}(x_k) \tag{1}$$

where $\mathbf{x}$ is $n_x$ system states; $\mathbf{A}_0$ is an $n_x \times n_x$ constant matrix; $\mathbf{b}$ is an $n_x \times 1$ constant vector, and $f_{\mathrm{FP}}$ is the freeplay function, reflecting the nonlinear connection between a displacement and a structurally restoring force. Assume that only one freeplay involved in the present work takes the $k$-th state $x_k$ as its input; then, the freeplay nonlinearity can be expressed as:

$$f_{\mathrm{FP}}(x_k) = \begin{cases} K_{\mathrm{lin}}(x_k - \delta) & x_k > \delta \\ 0 & |x_k| \le \delta \\ K_{\mathrm{lin}}(x_k + \delta) & x_k < -\delta \end{cases} \tag{2}$$

where $K_{\mathrm{lin}}$ is the underlying linear stiffness of the freeplay and $2\delta$ is the amount of freeplay. In the present paper, the $k$-th degree of freedom (DoF) of the system is called the "freeplay DoF" and $x_k$ is called the "freeplay state".

Equation (2) defines two non-smooth boundaries, which are known as the two "freeplay boundaries": $\Sigma_1 : x_k = -\delta$ and $\Sigma_2 : x_k = \delta$. These two boundaries separate the state

space into three subdomains: (1) $S_0$, where $|x_k| \leq \delta$; (2) $S_1$, where $x_k < -\delta$; and (3) $S_2$, where $x_k > \delta$. Within each subdomain, the system operates as a linear system, so the system is actually governed by two linear subsystems: (1) the underlying linear system (ULS), when the state $\mathbf{x}$ is in $S_0$, and (2) the overlying linear system (OLS), when $\mathbf{x}$ belongs to $S_1 \bigcup S_2$. Figure 1 illustrates freeplay and the three subdomains $S_{1,2,3}$, separated by two freeplay boundaries $\Sigma_{1,2}$ in state space.

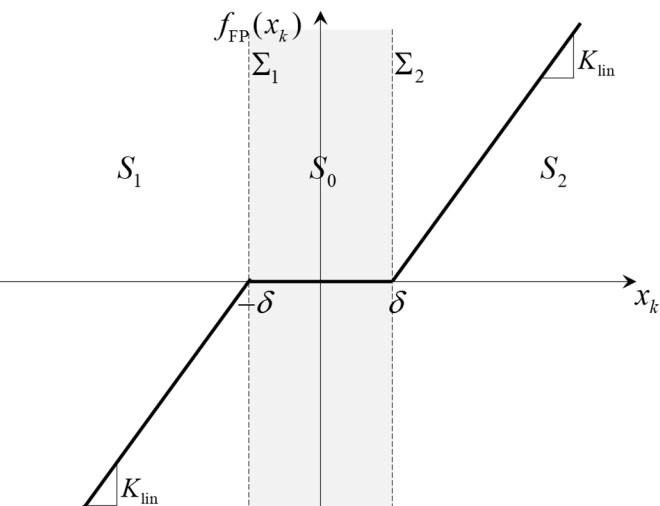

**Figure 1.** Illustration of freeplay and three subdomains separated by two freeplay boundaries in state space.

### 2.2. Time Integrations Based on the Runge–Kutta and Hénon Methods

Time integration using a combined fourth–fifth-order Runge–Kutta and the Hénon method (Hénon–RK45) was proposed by Hénon [7] and is widely adopted when solving a set of state space equations with non-smooth nonlinearities. For simplicity, the state space Equation (1) are expressed as $\dot{\mathbf{x}} = \mathbf{f}(\mathbf{x})$. The Hénon–RK45 method integrates the equations using an ordinary fourth–fifth-order Runge–Kutta scheme (RK45) initially until any freeplay boundary is crossed by $x_k(t)$. Assume that the latest time instance before crossing is $t_{\text{break}}$, and the states at that time are $\mathbf{x}_{\text{break}}$. Then, the Hénon method moves the freeplay state $x_k$ to the about-crossing boundary precisely via the following steps: (1) Change the integrating variable from time $t$ to the freeplay state $x_k$; that is,

$$\frac{\mathrm{d}}{\mathrm{d}x_k}\begin{bmatrix} \mathbf{x} \\ t \end{bmatrix} = \begin{bmatrix} \dfrac{\mathrm{d}\mathbf{x}}{\mathrm{d}t} \cdot \dfrac{\mathrm{d}t}{\mathrm{d}x_k} \\ 1 \cdot \dfrac{\mathrm{d}t}{\mathrm{d}x_k} \end{bmatrix} = \begin{bmatrix} \dot{\mathbf{x}}/\dot{x}_k \\ 1/\dot{x}_k \end{bmatrix} = \begin{bmatrix} \mathbf{f}(\mathbf{x})/\dot{x}_k \\ 1/\dot{x}_k \end{bmatrix} \tag{3}$$

Thus, a new set of state space equations can be constructed as below:

$$\frac{\mathrm{d}}{\mathrm{d}x_k}\mathbf{y} = \mathbf{f}^*(\mathbf{y}) \tag{4}$$

where $\mathbf{y} = [\mathbf{x}^T, t]^T$ is the new vector of the states. (2) Integrate Equation (4) using RK45 from $\mathbf{y}_0 = [\mathbf{x}_{\text{break}}^T, t_{\text{break}}]^T$ as $x_k$ is moved from $x_{k,\text{break}}$ to the about-crossing boundary,

where $\mathbf{y}$ becomes $\mathbf{y}_e = [\mathbf{x}_e^T, t_e]^T$. (3) Resume the integration of $\mathbf{x}(t)$ using the RK45 scheme from $\mathbf{x}_e(t_e)$ until the next crossing event occurs, and then repeat steps (1) to (3) until the system responses are obtained within a sufficiently long time range.

## 3. State Space Iterating (SSI) Scheme

### *3.1. Preliminary Concepts*

In this section, four important concepts are introduced before we propose the state space iterating (SSI) scheme: (1) the Poincaré map, (2) the Lorenz map, (3) the attractors, and (4) the basins of attraction. The first two concepts are introduced in Section 3.1.1, while the last two are provided in Section 3.1.2.

### 3.1.1. Poincaré Map and Lorenz Map

For a given initial condition (I.C.) $\mathbf{x}_0$, a start time $t_0$, and a time range $T$, Equation (1) determines a solution $\mathbf{x}(t)$, where $t_0 \le t \le t_0 + T$. Such a solution can be depicted as an orbit $\Gamma$ in state space, as shown in Figure 2a, where the closed orbit $\Gamma_C$ indicates a periodic solution of the system. The Poincaré map was defined by Poincaré in 1881 [15]. Perko provided good interpretations of the Poincaré map in Section 3.4 of his book [16]. Suppose that $\Sigma$ is a hyperplane perpendicular to a periodic orbit $\Gamma_C$ at $\mathbf{x}_0$; then, for any point $\mathbf{x} \in \Sigma$ sufficiently near $\mathbf{x}_0$, the orbit $\Gamma$ through $\mathbf{x}$ at $t = 0$ will cross $\Sigma$ again at another point $P(\mathbf{x})$. The mapping $\mathbf{x} \rightarrow P(\mathbf{x})$ is called a Poincaré map, or the first return map. Figure 2a illustrates the Poincaré map in state space.

If we could find a fixed point $\mathbf{x}_L$ on the Poincaré map—that is, $\mathbf{x}_L - P(\mathbf{x}_L) = 0$— then a periodic solution would be found at $\mathbf{x}_L$ and an LCO could be determined. However, only a few researchers have tried to find the fixed points of a Poincaré map. An example can be found in Monfared's paper [17], in which he derived the expression of a Poincaré map $P(\mathbf{x})$ numerically and calculated the fixed points of $P(\mathbf{x})$. Nevertheless, the results presented by Monfared were dependent on an artificial parameter $\lambda$ that has no physical meaning, as the author stressed. The sophisticated procedures for finding fixed points on a Poincaré map could be difficult to apply to a general aeroelastic system.

Other applications of Poincaré maps are actually based on phase diagrams, upon which a set of points at the Poincaré section $\Sigma$ are projected. This phase diagram consists of many discrete points and is usually simply called a "Poincaré section". Figure 2b shows a Poincaré section derived by Dimitriadis [13], based on which the author explained the mechanism of the associated bifurcations and confirmed that the system was likely undergoing chaotic motion. Many examples can be found in Figure 12 in [10], Figure 10 in [18], Figure 13 in [19], and Figure 11 in [20]. The Poincaré section provides evidence for the type of nonlinear response (e.g., LCO, quasi-periodic motion, or chaotic motion). However, it cannot aid in time integration for finding LCOs. Instead, it usually plays the role of confirming what has been derived from time integration.

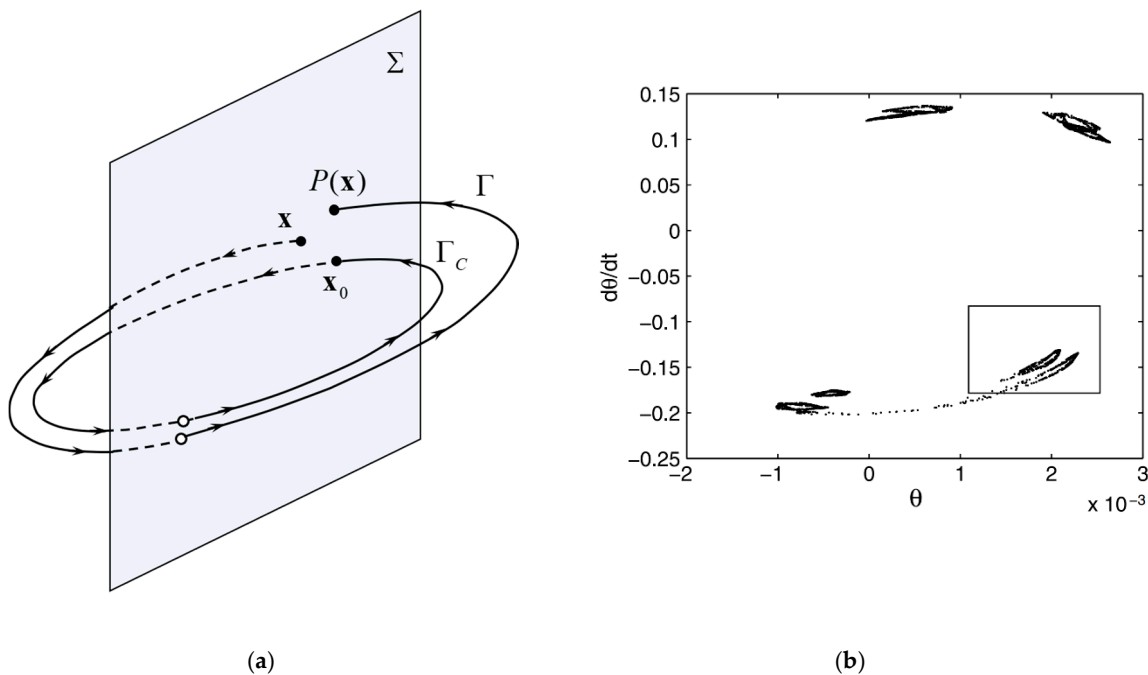

| (**a**) | (**b**) |

**Figure 2.** Illustration of a Poincaré map on (**a**) state space and (**b**) the Poincaré section. (**a**) is based on Figure 1 in Section 3.4 of Perko's book, and (**b**) is based on Figure 20 (**a**) in Dimitriadis' paper. (**b**) shows the normal method for a Poincaré map to be applied on the analysis of nonlinear behaviors in an aeroelastic system.

The Lorenz map was proposed by Lorenz [21] while studying his famous three-DoF system from a simplified model of convection rolls in the atmosphere. An excellent introduction on Lorenz's system and the Lorenz map is presented by Strongatz in Section 9 of his book [22]. In Lorenz's system, three system states, $x$, $y$, and $z$, vary with time. To grasp the features of the system, Lorenz focused only on the connection between the $n$-th local maximum $z_n$ and the next one, $z_{n+1}$, as shown in Figure 3. Then, a map $z_{n+1} = L(z_n)$, known as the Lorenz map, was obtained through numerous combinations of $z_n$ and $z_{n+1}$, as presented in Figure 4. Both Figures 3 and 4 were originally presented by Strogatz [22].

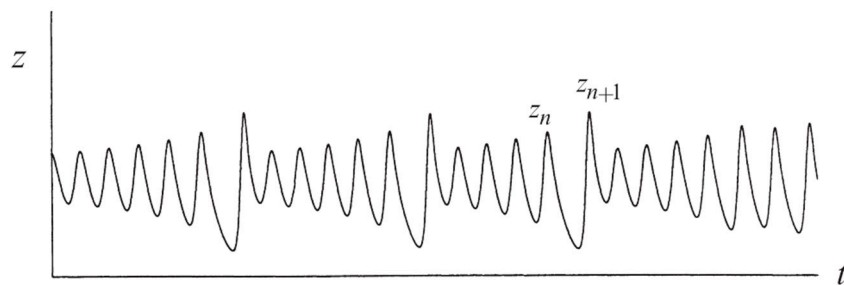

**Figure 3.** Lorenz focused on the connection of two adjacent local maxima in the system response. This figure is based on Figure 9.4.2 in Strogatz's book.

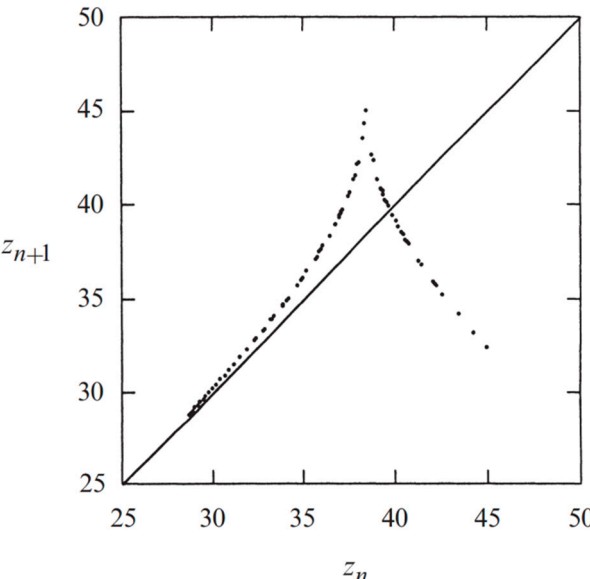

**Figure 4.** The Lorenz map reflects the feature of the nonlinear response in Figure 3 visually. This figure is based on Figure 9.4.3 in Strogatz's book.

The Lorenz map (Figure 4) is significant because it reflects the nonlinear pattern of $z(t)$. Note that there are almost no "thicknesses" to the graph in Figure 4, so the pattern of $z(t)$ is clear. For any given $z_0$, we can predict $z_1$ by $z_1 = L(z_0)$, followed by $z_2 = L(z_1) \dots$; thus, an iteration procedure can be constructed and allows us to predict $z_n$ simply by using a Lorenz map $n$ times [22]:

$$z_n = \underbrace{L(L(\cdots L(z_0)\cdots))}_{n} = L^n(z_0)$$

(5)

The fixed points of the Lorenz map, $z_L : z_L - L(z_L) = 0$, also determine the periodic solutions, that is, the LCOs, and one can find these fixed points via a visual inspection (e.g., finding the intersections of $z_{n+1} = L(z_n)$ and $z_{n+1} = z_n$ in Figure 4). The Lorenz map also provides a visual approach to determine the stability of $z(t)$. Lorenz noted that $z(t)$ is always unstable since $|L(z_n)| > 1$ can be seen at every point in Figure 4.

Since Figure 4 was obtained at a series of discrete points, the "worst-case" scenario of $z_0$ could disobey the pattern of $L$ shown in Figure 4. Tucker [23,24] proved that the Lorenz map does accurately reflect the real features of the system and that the Lorenz attractor exists. Stewart [25] and Viana [26] provided some interpretations of Lorenz's work and stressed the significance of attractors in the Lorenz map. These attractors will be introduced in the next section.

### 3.1.2. Attractor and Basin of Attraction

The attractor is a special set where mapping $F$ shows the attracting features and is helpful for analyzing nonlinear behaviors. For a system with $n_x$ states $\mathbf{x}(t)$, Strogatz defined an attractor as a set $U_{min} \subset \mathbb{R}^{n_x}$ of the mapping $F : \mathbf{x}_{n+1} = F(\mathbf{x}_n)$ as follows (see Section 9.3 in [22]): (1) Any $\mathbf{x}(t)$ that starts in $U_{min}$ stays in $U_{min}$ for all time; (2) there exists an open set $U$ containing $U_{min}$ such that if $\mathbf{x}_0 \in U$, then the distance

from $\mathbf{x}(t)$ to $U$ tends toward zero as $t \rightarrow \infty$; and (3) there is no proper subset of $U_{\min}$ that satisfies both (1) and (2).

In summary, a set $U_{\min}$ is an attractor of a mapping $F: \mathbf{x}_{n+1} = F(\mathbf{x}_n)$ if

$$\begin{cases} 1. & \forall \mathbf{x}_0 \in U_{\min} : \ \mathbf{x}_1 \in U_{\min} \\ 2. & \exists U \supseteq U_{\min} \text{ and } \forall \mathbf{x}_0 \in U : \ \lim_{n \to \infty} F^n(\mathbf{x}_0) = \mathbf{x}_n \in U_{\min} \\ 3. & \text{there is no } U_{\text{sub}} \subset U_{\min} \text{ satisfies 1. and 2.} \end{cases} \tag{6}$$

where $U$ is called the "basin of attraction" for the attractor $U_{\min}$.

An LCO is actually a special attractor that contains only one point, a fixed point $\mathbf{x}_L$, which allows

$$\mathbf{x}_L - F(\mathbf{x}_L) = \mathbf{0} \tag{7}$$

Note that the mapping $F$ could be a Poincaré map, Lorenz map, or any other carefully designed type of mapping which reflects the periodicity of the system when any LCO occurs.

*3.2. Basic Ideas of SSI*

As explained in Section 3.1.2, one can find an LCO solution for a system by finding the fixed point of a mapping $F$. Suppose that the system has $n_L$ LCOs. Since each LCO $\mathbf{x}_{L,j}$ is an attractor and is associated with a basin of attraction $U_{L,j}$, according to Equation (6), we could define a set $X_L$ that collects all LCOs and a set $\bar{U}_L$ that contains all basins of attraction—that is, $X_L = \{\mathbf{x}_L \mid \mathbf{x}_L - F(\mathbf{x}_L) = \mathbf{0}\}$ and $\bar{U}_L = \bigcup_{j=1}^{n_L} U_{L,j}$. Our final goal is to find $X_L$ and distinguish each separate point $\mathbf{x}_L$ in $X_L$.

Two basic conceptions thus emerge: (1) If we use the Poincaré map $P$ as $F$, there is no method for finding fixed points easily; if we use a Lorenz map $L$ as $F$, although it is easy to find fixed points via a visual inspection of Figure 4, the results highly depend on the accuracy of the local maxima. Since we have already used the Hénon–RK45 method to move system states accurately into discontinuous boundaries, we expect to find a way to use the states that we already have instead of spending additional and considerable time to find local maxima. Therefore, we need to construct a new mapping $F$ instead of using $P$ or $L$ directly.

(2) If the set $X_L$ is extremely difficult to obtain, we can initially find the basins of attraction $\bar{U}_L$ because we know that $X_L \subseteq \bar{U}_L$ according to Equation (6). Moreover, if $\bar{U}_L$ is still too difficult obtain, then we can try to find an iteration in state space:

$$\bar{U}_L \supseteq \mathbf{X}_0^{\{1\}} \supseteq \mathbf{X}_0^{\{2\}} \supseteq \cdots \supseteq \mathbf{X}_0^{\{n\}} \supseteq X_L \tag{8}$$

We can start with any $\mathbf{X}_0^{\{j\}}$ that could possibly be obtained, confirm that $\bar{U}_L \supseteq \mathbf{X}_0^{\{j\}} \supseteq X_L$, and then try to contract $\mathbf{X}_0^{\{j\}}$ by finding another set $\mathbf{X}_0^{\{j+1\}}$ which would satisfy $\bar{U}_L \supseteq \mathbf{X}_0^{\{j\}} \supseteq \mathbf{X}_0^{\{j+1\}} \supseteq X_L$. Thus, an iteration $\mathbf{X}_0^{\{j\}} \rightarrow \mathbf{X}_0^{\{j+1\}}$ in state space needs to be constructed, which may need some additional results from time integration. Finally, when we find $X_L$ and the system patterns are clear enough, we can read all $\mathbf{x}_L$ values easily from a figure which is similar to Figure 4.

Thus, we have two goals: (1) design a mapping $F$ and (2) construct the iteration procedure shown in Equation (8), which will be implemented in Sections 3.3 and 3.4, respectively.

### 3.3. Modification of the Poincaré Map and Iteration Plots

First, an $N_T$-th time return mapping $F = F^{N_T}$ is proposed based on the Poincaré map. We specify a Poincaré section $\Sigma$ at the upper freeplay boundary, $\Sigma : x = \delta$; then, let the system evolve from a point $\mathbf{x}_0(t_0)$ at $\Sigma$ to another point $\mathbf{x}_0^*(t_0^*)$ at $\Sigma$ after $N_T$ times crossing $\Sigma$, where $N_T$ is the "crossing number" and can be specified as any positive integer. The orbit of $\mathbf{x}(t)$, which represents the evolution, is obtained by time integration with the Hénon–RK45 method and is illustrated in Figure 5a. The proposition of $N_T$ is based on the consideration that one may encounter a lengthy transient phase of the system responses due to an inappropriate selection of $\mathbf{x}_0$ and the complexity of the aeroelastic system with freeplay. Figure 5b provides a general sense of the freeplay state $x_k$ when the system is evolving from $\mathbf{x}_0(t_0)$ to $\mathbf{x}_0^*(t_0^*)$ following the orbit $\Gamma$.

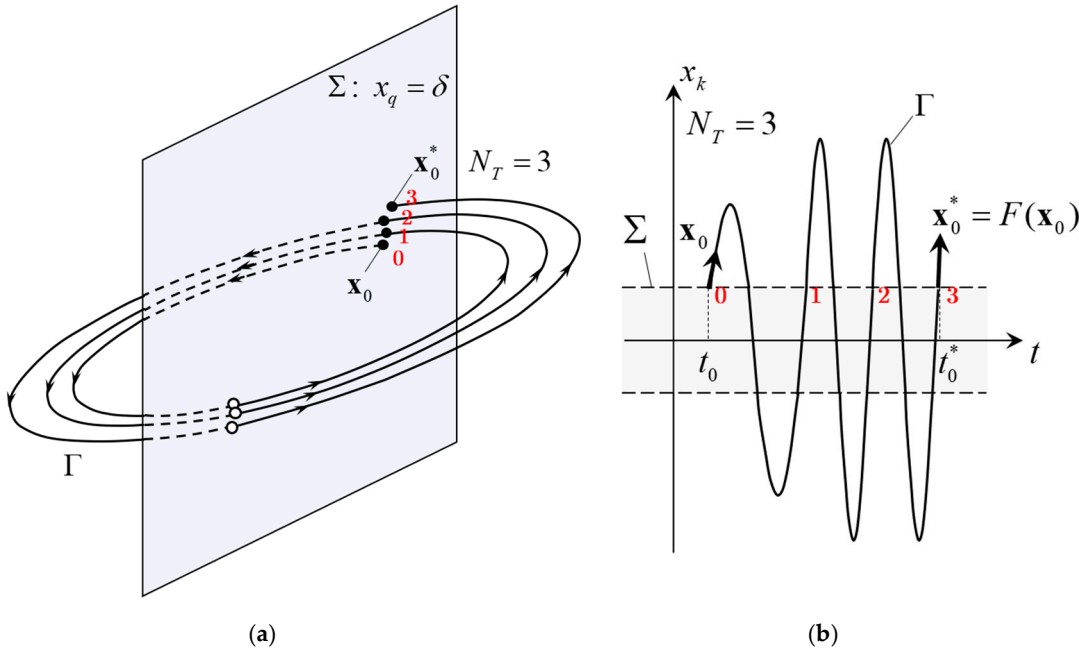

(**a**)　　　　　　　　　　　　　　　　　　　　　　　　　　　　　(**b**)

**Figure 5.** Illustration of the $N_T$-th time return mapping. (**a**) is depicted in state space and (**b**) is depicted on the plane of the freeplay state vs. time. The parameter $N_T = 3$ is given as an example.

Next, we will observe the orbit $\Gamma$ from another perspective and focus on the $j$-th state $x_j$. When the integration starts with $\mathbf{x}_0(t_0)$, we have $x_j(t_0) = x_{0,j}$, while when the integration is terminated at $\mathbf{x}_0^*(t_0^*)$, we have $x_{0,j}^*(t_0^*) = x_{0,j}^*$. Inspired by the Lorenz map, we next construct a two-DoF plane, as shown in Figure 6, where point $A$ represents the orbit $\Gamma$ by means of its coordinates $(x_{0,j}, x_{0,j}^*)$. This is similar to what Lorenz did in Figure 4, where Lorenz used a point with the coordinates $(z_{n+1}, z_n)$ to represent a segment of system behaviors. By means of this method, each I.C. case $\mathbf{x}_0$ is associated with $\mathbf{x}_0^*$ and a point on the two-DoF plane. If we select $N_S$ I.C. cases, then we will have

$N_S$ points on the plane. When $N_S$ is a large number, sufficient points on the plane could illustrate a spatial pattern, indicating system behaviors. Based on this pattern, we can construct the mapping $F^{N_T}$: $\mathbf{x}_0^* = F^{N_T}(\mathbf{x}_0)$. We call this two-DoF plane an "iteration plot". Note that we need to construct $n_x - 1$ iteration plots (where $n_x$ is the number of system states), with each one corresponding to a state except for the freeplay state $x_k$, because $x_k$ always equals $\delta$ since both $\mathbf{x}_0$ and $\mathbf{x}_0^*$ are at $\Sigma$.

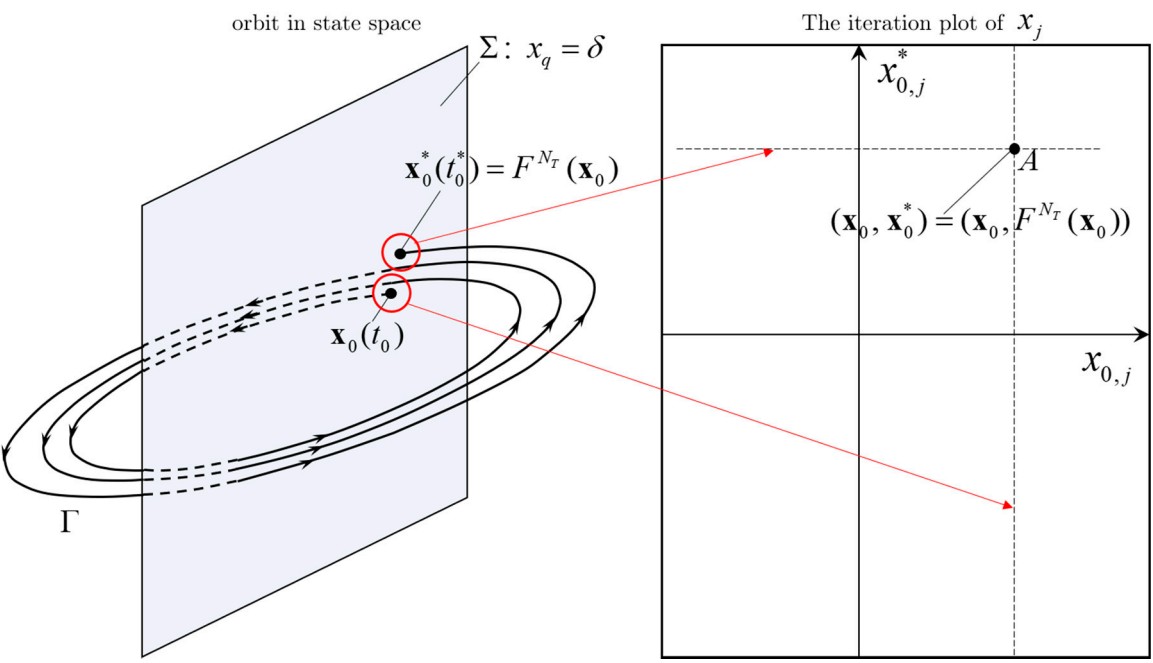

**Figure 6.** Illustration of an iteration plot, in which point *A* represents how a state behaves when the system evolves from the upper freeplay boundary and returns to that boundary after crossing it several times.

### 3.4. LCO Anlaysis Using the SSI Scheme

As explained in Section 3.2, an LCO is a fixed point $\mathbf{x}_L$ of the mapping $F^{N_T}$: $\mathbf{x}_L = F^{N_T}(\mathbf{x}_L)$, from which we can find the fixed point simply by visual inspection. Add a line $l_{\text{equ}}$ in the iteration plot for $x_j$, which has a unit slope and indicates that $l_{\text{equ}}$: $x_{0,j}^* = x_{0,j}$, as shown in Figure 7a,b. Assume that $N_S$ points on the iteration plot form a perfect line $l_0$, which intersects $l_{\text{equ}}$ at Point $A$; then, $A$ provides the $j$-th component of $\mathbf{x}_L$: $x_{L,j} = x_{0,j}|_A = x_{0,j}^*|_A$. If a fixed point can be found at all iteration plots, then at those fixed points, all system states will behave as periodic motions, and an LCO is determined.

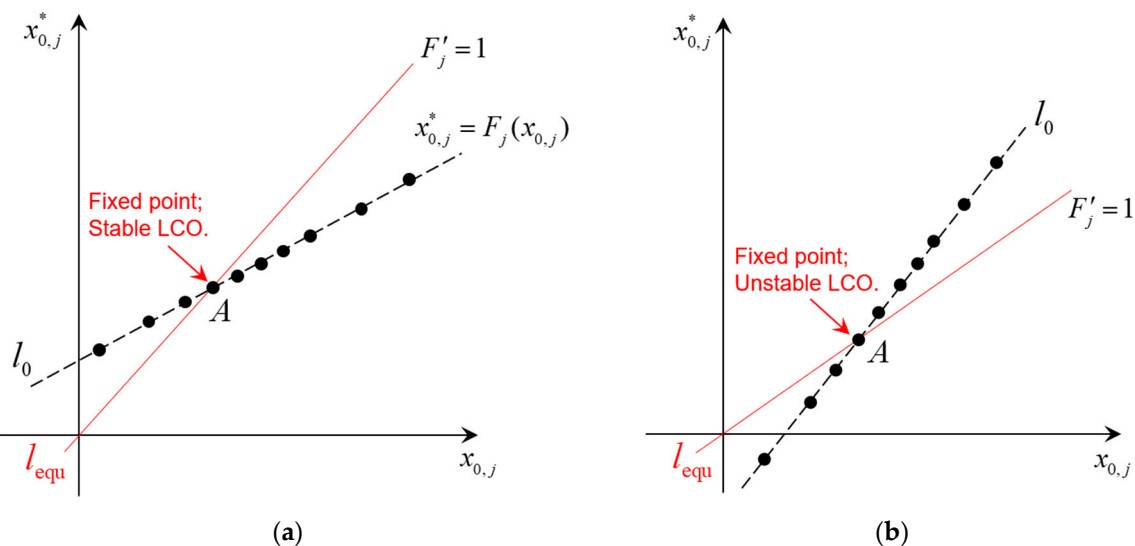

**Figure 7.** Illustration of the *j*-th iteration plot, where the spatial pattern indicates the occurrence of a limit cycle oscillation (LCO). The LCO is (**a**) stable or (**b**) unstable with respect to any perturbation in the *j*-th state.

Stability can also be determined visually. For simplicity, we use $F$ to represent the mapping instead of $F^{N_T}$. We define the slope $F_j'$ as $F_j' = dF_j / dx_{0,j}$, where $F_j$ is the $j$-th component of the mapping $x_{0,j}^* = F_j(x_{0,j})$. An LCO is unstable if $\left| F_j' \right| > 1$. We can prove this by linearizing $F_j$ near a fixed point. Suppose that the fixed point is at $x_{0,j} = x_{0,j}^* = \overline{x}$. A small perturbation $\delta x_{0,j} = x_{0,j} - \overline{x}$ is given as the input of the mapping. Then,

$$x_{0,j}^* = F_j(x_{0,j}) \approx F_j(\overline{x}) + F_j'(\overline{x}) \cdot (\overline{x} - x_{0,j}) \tag{9}$$

which means that

$$F_j(x_{0,j}) - F_j(\overline{x}) \approx F_j'(\overline{x}) \cdot (\overline{x} - x_{0,j}) \tag{10}$$

or

$$\left| \delta F_j \right| \approx \left| F_j'(x_{0,j}) \right| \cdot \left| \delta x_{0,j} \right| \tag{11}$$

As time progresses, the mapping will be repeated over and over again as $x_{0,j}^* = F_j(F_j(... F_j(x_{0,j}) ...))$, finally yielding:

$$\lim_{t \to \infty} \left| \delta F_j \right| = \lim_{N \to \infty} \left| F_j'(x_{0,j}) \right|^N \cdot \left| \delta x_{0,j} \right| \to +\infty \tag{12}$$

which indicates that the LCO is unstable. Conversely, if $\left| F_j' \right| < 1$, the LCO is stable. Figure 7 illustrates situations of both a stable and an unstable LCO. For simplicity, only one $l_{equ}$ representing $F_j' = 1$ is plotted.

However, to find fixed points may not be easy, since the patterns shown in the iteration plots may be complicated. As shown in Figure 8, the $N_S$ points form a range $R_0$ instead of the $l_0$ in Figure 7; therefore, there is no obvious fixed point that can be found. Presumably, a fixed point is hidden inside of the intersection line $L_0$: $L_0 = R_0 \cap l_{equ}$.

Here, we introduce an iterating procedure to gradually find the fixed point hidden in $L_0$ , which is the SSI scheme proposed in this paper.

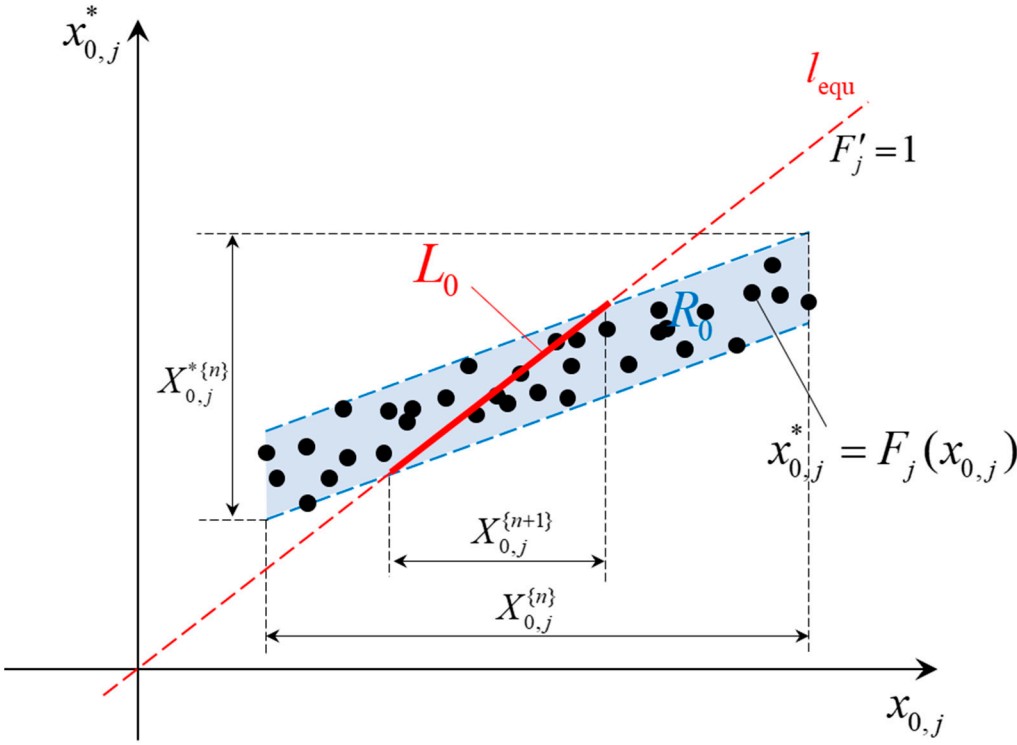

**Figure 8.** Illustration of the *j*-th iteration plot, where the spatial pattern indicates the occurrence of an LCO but the effects of transient phases and noises are also taken into consideration, so that the distributing region of all points is a belt region $R_0$ rather than a line, as shown in Figure 3.

First, for each state $x_j$, specify a range $X_{0,j}^{\{n\}}$ and select $N_S$ samples $x_{0,j}$ randomly from that range ( $x_{0,j} \in X_{0,j}^{\{n\}}$ ). By combining all $X_{0,j}^{\{n\}}$ values, we obtain a set $\mathbf{X}_0^{\{n\}} \in \mathbb{R}^{n_x - 1}$ that contains all $N_S$ I.C. cases, $\mathbf{x}_0 \in \mathbf{X}_0^{\{n\}}$. Carry out time integrations so that $\mathbf{x}_0^* = F^{N_T}(\mathbf{x}_0)$ can be obtained. Here, we need to check if all $\mathbf{x}_0^*$ values still remain in $\mathbf{X}_0^{\{n\}}$: $\mathbf{x}_0^* \in \mathbf{X}_0^{\{n\}}$ because we need to make sure that $\mathbf{X}_0^{\{n\}}$ is a subset of basins of attraction $\bar{U}_L$, as explained in Section 3.1.2 and Section 3.2—if not, expand the range of $\mathbf{X}_0^{\{n\}}$ until any $\mathbf{x}_0 \in \mathbf{X}_0^{\{n\}}$ leads to $\mathbf{x}_0^* = F^{N_T}(\mathbf{x}_0) \in \mathbf{X}_0^{\{n\}}$.

Second, construct iteration plots using $\mathbf{X}_0$ and $\mathbf{x}_0^*$ as explained in Section 3.4, where $N_S$ points form a region $R_0$ and intersect with $l_{equ}$ at $L_0$, as shown in Figure 8. The key point is that $L_0$ normally occupies a smaller range $X_{0,j}^{\{n+1\}}$ on $x_{0,j}$ than $X_{0,j}^{\{n\}}$ —that is, $X_{0,j}^{\{n+1\}} \subseteq X_{0,j}^{\{n\}}$. Thus, the range of $X_{0,j}^{\{n\}}$ can be contracted to $X_{0,j}^{\{n+1\}}$. By combining all $X_{0,j}^{\{n+1\}}$ on the iteration plots, a new set $\mathbf{X}_0^{\{n+1\}} \subseteq \mathbf{X}_0^{\{n\}} \subseteq \bar{U}_L$ is found.

Repeat the two procedures above, and a series set $\mathbf{X}_0^{\{1\}}$, $\mathbf{X}_0^{\{2\}}$, …, $\mathbf{X}_0^{\{n\}}$ can be obtained. The iteration will stop once $\mathbf{X}_0^{\{n\}}$ can no longer been contracted—that is, $\mathbf{X}_0^{\{n+1\}} \equiv \mathbf{X}_0^{\{n\}}$. After that, if $R_0$ no longer has a "thickness" to the graph, as shown in Figures 4 and 7, the $L_0$ in Figure 8 will degenerate to a fixed point $\mathbf{x}_L$, where an LCO

solution is found; otherwise, quasi-periodic motion or chaotic motion should be suspected in the set $\{\mathbf{x} \mid \mathbf{x} \in L_0 = R_0 \cap l_{\text{equ}}\}$. The present paper only focuses on LCOs, so any other kind of motion will be estimated as an LCO with a perturbation on the orbit. Studies on LCOs together with all other kinds of nonlinear responses are expected in our future works.

Note that when $N_T = 1$, the proposed mapping is a Poincaré map. However, we found that using $N_T > 1$ could accelerate the determination of clear patterns on the iteration plots and make the SSI scheme more robust. We tried using $N_T =$ 3, 5, 9, and 11, all of which yielded the same LCO results but required different numbers of SSI iterations to contract the set $\mathbf{X}_0^{\{n\}}$.

### 4. Numerical Model and Results of the Hénon–RK45 Method

In this section, a plunge–pitch wing section is introduced, as shown in Figure 9. Time integration with the Hénon–RK45 method is used to obtain the nonlinear LCO behaviors when symmetrical and non-preloaded freeplay nonlinearity is added to the pitch DoF of the wing section. The amount of freeplay is $2\delta = 0.1°$. The results in this section will serve as the benchmark for the next section, where the SSI scheme is applied to the same numerical model. The parameters of the wing section are listed in Table A1 in Appendix A.

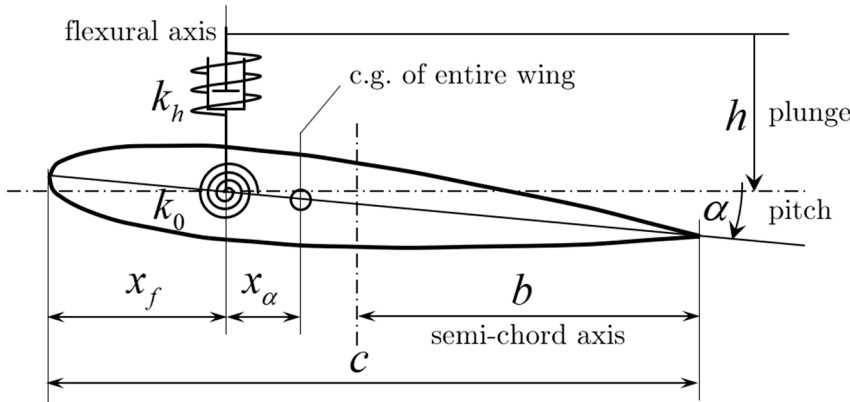

**Figure 9.** The numerical model: a plunge–pitch wing section, where freeplay nonlinearity is introduced in the pitch degree of freedom (DoF).

An unsteady aerodynamic model is established in the frequency domain following Theodorsen's method [27] and can be expressed as $\mathbf{f}_a(k) = \frac{1}{2}\rho U^2 \mathbf{Q}\mathbf{q}(k)$, where $\rho = 1.225$ kg/m³ is the air density, $k$ is the reduced frequency, $U$ is the flow velocity (m/s), $\mathbf{Q}$ is the aerodynamic influence coefficient (AIC) matrix, and $\mathbf{q} = [h, \alpha]^T$ are the generalized modal coordinates, where $h$ and $\alpha$ denote the plunge (m) and pitch angle (rad), respectively. Then, the rational function approximation (RFA) introduced by Roger [28] and developed by Karpel [29] and the minimal state (MS) method proposed by Sherwood and Karpel [30] are used to approximate the aerodynamic loads in the time domain, $\tilde{\mathbf{f}}_a(t)$. The equations of motion for the wing section model are as follows:

$$(\mathbf{M}_S\ddot{\mathbf{q}} + \mathbf{D}_S\dot{\mathbf{q}} + \mathbf{K}_{S1}\mathbf{q})/L = \tilde{\mathbf{f}}_a(t) = \tfrac{1}{2}\rho U^2 \mathbf{A}_0\mathbf{q} + \tfrac{1}{2}\rho b U \mathbf{A}_1\dot{\mathbf{q}} + \tfrac{1}{2}\rho b^2 \mathbf{A}_2\ddot{\mathbf{q}} + \tfrac{1}{2}\rho U^2 \mathbf{A}_D\mathbf{r}$$
$$\dot{\mathbf{r}} = \mathbf{A}_E\dot{\mathbf{q}} + U/b\,\mathbf{R} \tag{13}$$

where $\mathbf{M}_S$, $\mathbf{D}_S$, and $\mathbf{K}_{S1}$ represent the structural mass, damping, and stiffness, respectively; $\mathbf{A}_0$, $\mathbf{A}_1$, $\mathbf{A}_2$, $\mathbf{A}_D$, and $\mathbf{A}_E$ are matrices of the aerodynamic coefficients; and

$\mathbf{R}$ is a diagonal matrix that contains two lag terms, $r_1 = -0.08$ and $r_2 = -0.60$. The accuracy of $\tilde{\mathbf{f}}_a(t)$ approximating $\mathbf{f}_a(k)$ in Equation (13) is examined by converting $\tilde{\mathbf{f}}_a(t)$ in the Laplace domain, where $L(\tilde{\mathbf{f}}_a(t)) = \frac{1}{2}\rho U^2 \tilde{\mathbf{Q}}(s)\mathbf{q}(s)$. Next, let the Laplace operator be $s = ik$, where $i$ is an imaginary unit. Figure 10 shows a comparison between $\tilde{\mathbf{Q}}(ik)$ and $\mathbf{Q}(k)$ with 18 reduced frequencies: $k$ = 0.005, 0.01, 0.02, 0.03, 0.04, 0.05, 0.06, 0.08, 0.10, 0.12, 0.15, 0.20, 0.25, 0.30, 0.35, 0.40, 0.45, and 0.50. This result proves that the aerodynamic model in Equation (13) is qualified for further calculations.

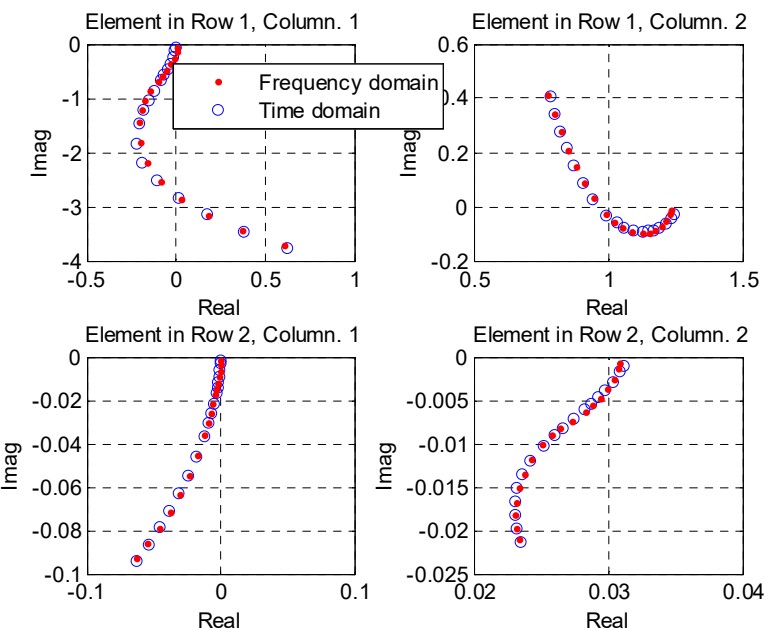

**Figure 10.** The aerodynamic influence coefficient (AIC) matrix obtained in the frequency domain and approximated in the time domain via the rational function approximation (RFA) and the minimal state (MS) methods. Two lag terms, −0.08 and −0.06, are included in the RFA.

When freeplay nonlinearity is added to the pitch DoF, the linear structurally restoring forces $\mathbf{K}_{Sl}\mathbf{q}$ will be replaced by $\mathbf{K}_{S0}\mathbf{q}$. Equations of motion for the wing section with freeplay can be expressed in the form of a set of piecewise linear state space equations:

$$\dot{\mathbf{x}}(t) = \mathbf{A}_{lin}\mathbf{x}(t) + \mathbf{b}K_\alpha f_{NL}(\alpha) \tag{14}$$

where $\mathbf{x} = [\mathbf{q}^T, \dot{\mathbf{q}}^T, \mathbf{r}^T]^T$ represents the system states; $K_\alpha$ is the linear stiffness of pitch DoF; $\mathbf{A}_{lin}$ represents the linear coefficient matrices for the ULS; $\mathbf{b}$ is a constant vector; and the eigenvalues of $\mathbf{A}_{lin}$ are shown in Figure 11. For a clear comparison, the eigenvalues of matrix $\mathbf{A}_{NL}$ of the OLS are presented in Figure 12. The parameters involved in Equations (13) and (14) are presented in Appendix A.

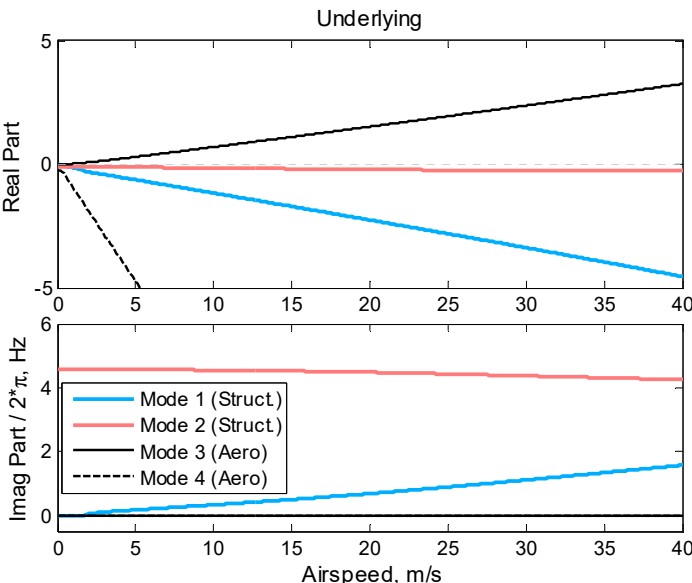

**Figure 11.** Eigenvalues of the underlying linear system (ULS) of the wing section model varying with the flow velocity.

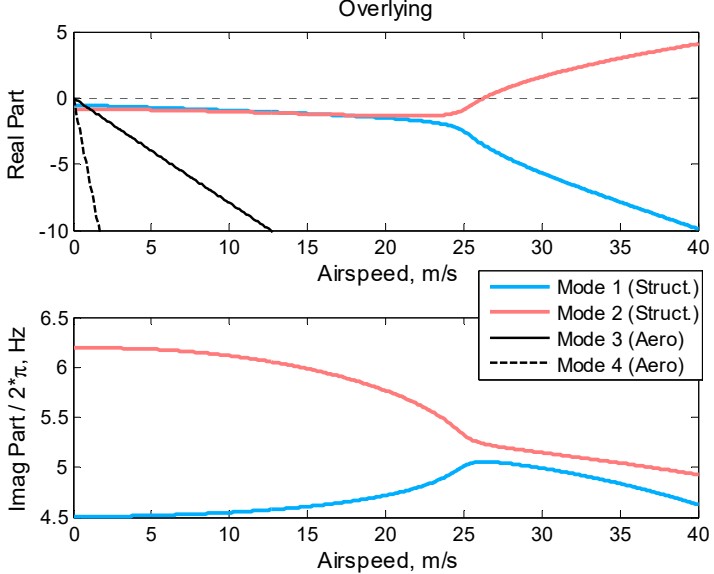

**Figure 12.** Eigenvalues of the overlying linear system (OLS) of the wing section model varying with the flow velocity.

## 5. LCO Results of Time Integrations with the Hénon–RK45 Method

An I.C. case $\mathbf{x}_0 = \mathbf{x}(t)$ is specified as a case where only the pitch angle $\alpha(t)$ starts with a non-zero value $\alpha_0$, while the rest of the system states are zero—that is, $\mathbf{x}_0 = [0, \alpha_0, 0, 0, 0, 0]^T$. The amount of freeplay is $2\delta$ = 0.1°. We first carried out time integrations with the Hénon–RK45 method at an airspeed of $U$ = 20 m/s and various $\alpha_0$ values. The results show that the system behaviors depend highly on $\alpha_0$. For example, when the initial non-dimensional pitch angle, $\alpha_0/\delta$, equals 5.00, a three-domain LCO (3-D LCO) is found in which the phase trajectory of $\alpha(t)$ between 13 and 15 s passes all three subdomains defined in Section 2.1, as illustrated in Figure 13b. However, when

$\alpha_0/\delta$ = 1.00, a two-domain LCO (2-D LCO) emerges, and its phase trajectory passes only two subdomains, $S_0$ and $S_2$, as presented in Figure 14.

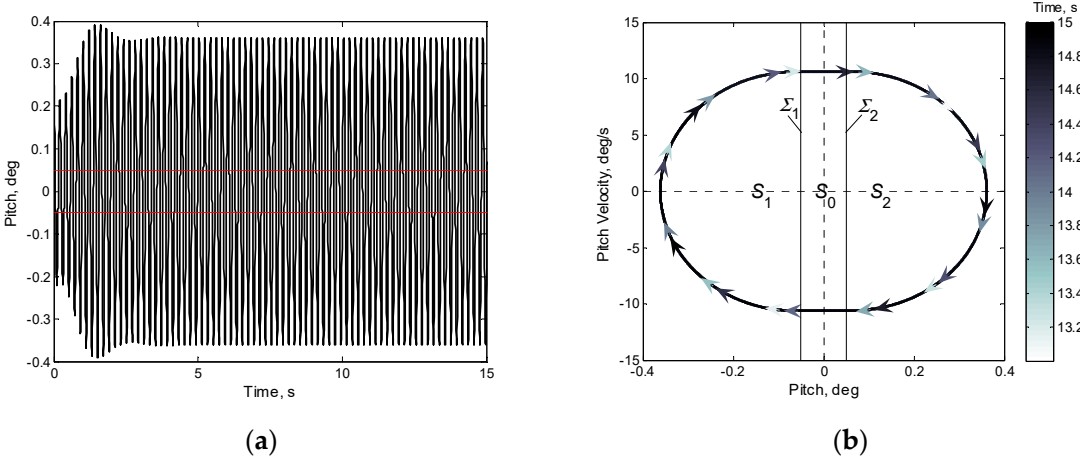

**Figure 13.** A three-domain LCO (3-D LCO) emerges when the flow velocity is 20 m/s and the initial non-dimensional pitch angle is 5.00, where the response of the freeplay state is presented by (**a**) a time history of 0–15 s and (**b**) a phase trajectory of 13–15 s.

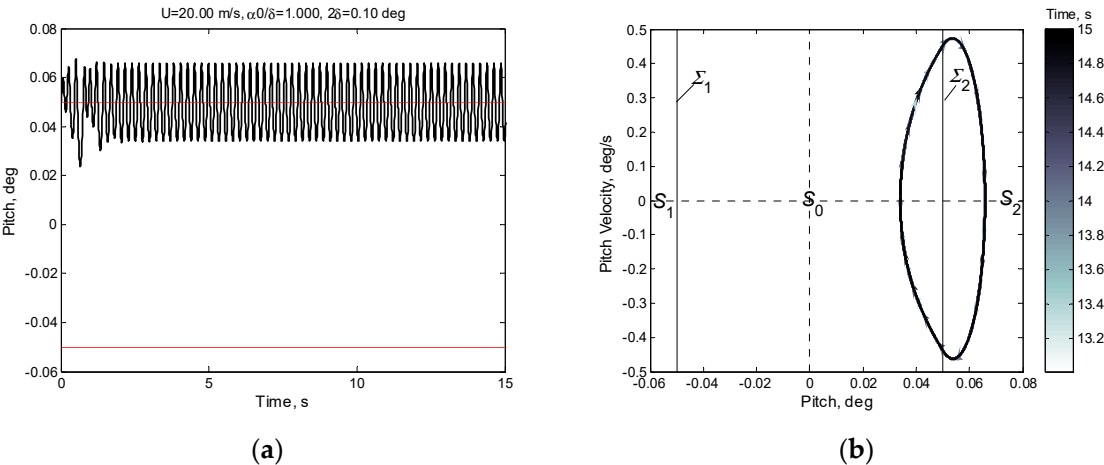

**Figure 14.** A two-domain LCO (2-D LCO) emerges when the flow velocity is 20 m/s and the initial non-dimensional pitch angle is 1.00, where the response of the freeplay state is presented by (**a**) a time history of 0–15 s and (**b**) a phase trajectory of 13–15 s.

To determine the effect of $\alpha_0/\delta$ on LCO behaviors, 100 cases of $\alpha_0/\delta$ that vary from 0.1 to 100 subjects with a 1-cos function and 73 cases of airspeed $U$ that increases from 12 to 20 m/s following a piecewise linear function were considered, as illustrated in Figure 15. Therefore, 7300 cases featuring a combination of $\alpha_0/\delta$ and $U$ were specified, and the results derived from the time integrations are presented in Figure 16.

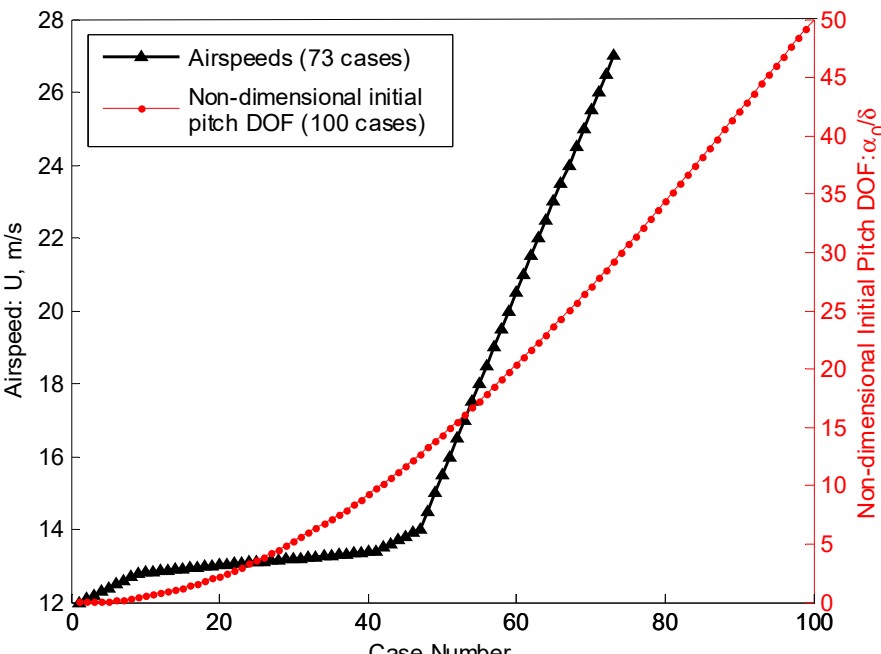

**Figure 15.** Calculation cases with 73 airspeeds and 100 initial pitch angles for each airspeed.

It was found that the 3-D LCO exists in a continuous region with a clear boundary, while the 2-D LCOs are distributed in numerous separated regions, as shown in Figure 16. A close-up of the boundary of the 3-D and the 2-D LCO regions is provided in Figure 17. Since Figures 16 and 17 can only be obtained under various $\alpha_0$ and $U$ values, we are still unaware about how other states will affect the two LCO regions if they have non-zero values in the I.C. cases.

The time integrations for the 7300 cases consume a total central processing unit (CPU) time of 4687 s (about 1.3 h); however, only the effect of $\alpha_0$ is investigated among all system states. All calculations in this paper were carried out via the software package MATLAB r2014 running on a computer with a quad-core and an eight-thread processor (Intel Core i5-10210U CPU @ 1.60 GHz and 2.11 GHz) and 16.0 GB of random access memory.

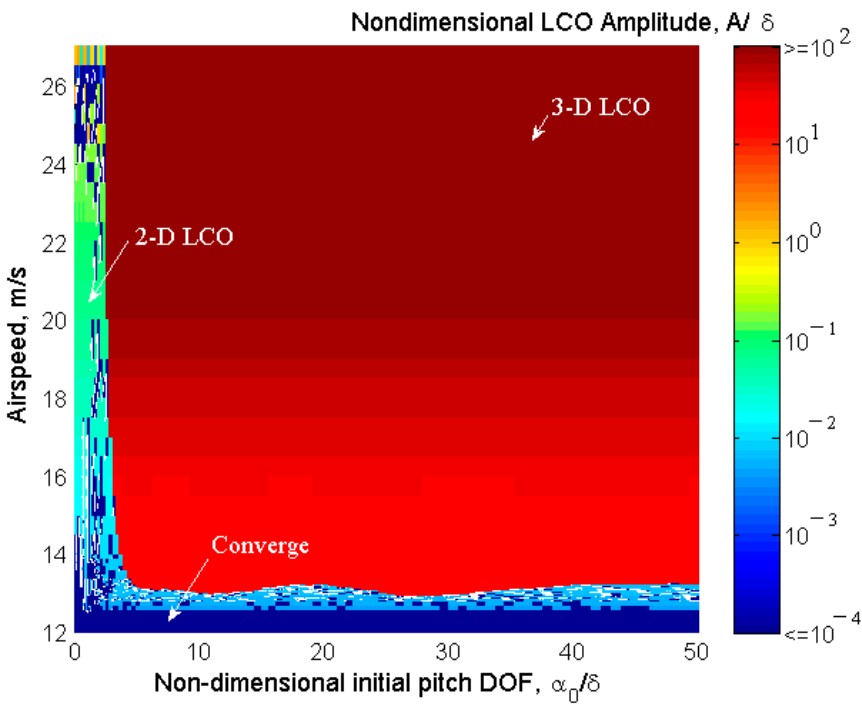

**Figure 16.** System behaviors derived from the 7300 cases, where a 3-D LCO region with a clear boundary and a 2-D LCO region mingled with the converging region are found.

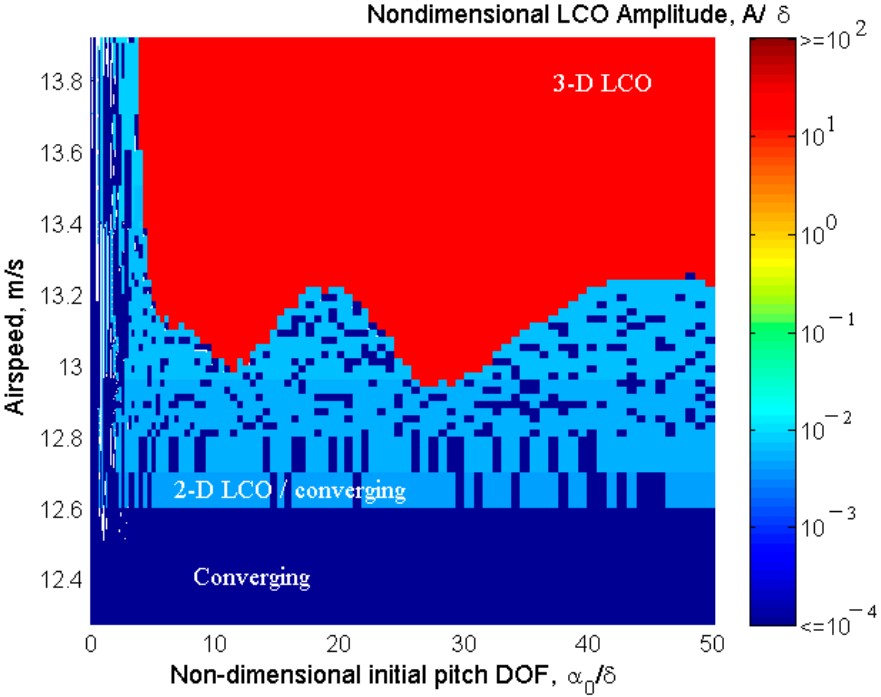

**Figure 17.** A close-up of Figure 16 which focuses on the airspeed from 12.4 to 13.8 m/s. Random switching between the 2-D LCO and convergence of the system responses can be observed as the initial pitch angle increases.

## 6. LCO Analysis of Time Integrations with the SSI Scheme

### 6.1. LCO Results under an Airspeed of 20 m/s

In this section, we seek to find all possible LCO solutions using time integration with the proposed SSI scheme. The SSI scheme is introduced in Section 3. First, an airspeed of

$U$ = 20 m/s and the amount of freeplay $2\delta$ = 0.1° are prescribed. A Poincaré section is specified as $\Sigma: x_2 = \alpha = \delta$, and the crossing of system states $\mathbf{x}(t)$ to $\Sigma$ is valid only if the crossing direction follows $x_4 = \dot{\alpha} > 0$. Second, $N_T$-time return mapping $F$: $\mathbf{x}_0^{\{n+1\}} = F(\mathbf{x}_0^{\{n\}})$ is introduced with $N_T$ = 11, where $\mathbf{x}_0^{\{n\}}$ and $\mathbf{x}_0^{\{n+1\}}$ are both at $\Sigma$. Our final goal is to find all fixed points $\mathbf{x}_L$ of $F$ that actually represent the LCO solutions of the system. To do so, a set $\mathbf{X}_0^{\{1\}} \subset \mathbb{R}^{n_x - 1}$ is specified as follows:

$$\mathbf{X}_0^{\{1\}} = \{\mathbf{x}_0 \mid \mathbf{x}_0 \ s.t. \begin{cases} -0.1 \leq & x_{0,1} = h_0 & \leq 0.1 \\ & x_{0,2} = \alpha_0 & \equiv \delta \\ -0.1 \leq & x_{0,3} = \dot{h}_0 & \leq 0.1 \\ 0 < & x_{0,4} = \dot{\alpha}_0 & \leq 30.0 \\ -0.1 \leq & x_{0,5} = r_1 & \leq 0.1 \\ -0.1 \leq & x_{0,6} = r_2 & \leq 0.1 \end{cases} \} \tag{15}$$

$N_S$ = 1000 samples of $\mathbf{x}_0^{\{1\}} \in \mathbf{X}_0^{\{1\}}$ are selected randomly using the Latin Hypercube Sampling (LHS) algorithm. Then, $N_S$ states $\mathbf{x}_0^{\{2\}}$ are calculated by mapping $\mathbf{x}_0^{\{2\}} = F(\mathbf{x}_0^{\{1\}})$ using time integrations, and $N_S$ points based on $\mathbf{x}_0^{\{2\}}$ and $\mathbf{x}_0^{\{1\}}$ are plotted on $n_x - 1$ = 5 iteration plots, as explained in Sections 3.3 and 3.4. The iteration plot for the plunge, $x_1$, is presented in Figures 18 and 19 provides an interpretation of Figure 18 to visualize the connection between the mapping $F$ in state space and the iteration plot for $x_1$.

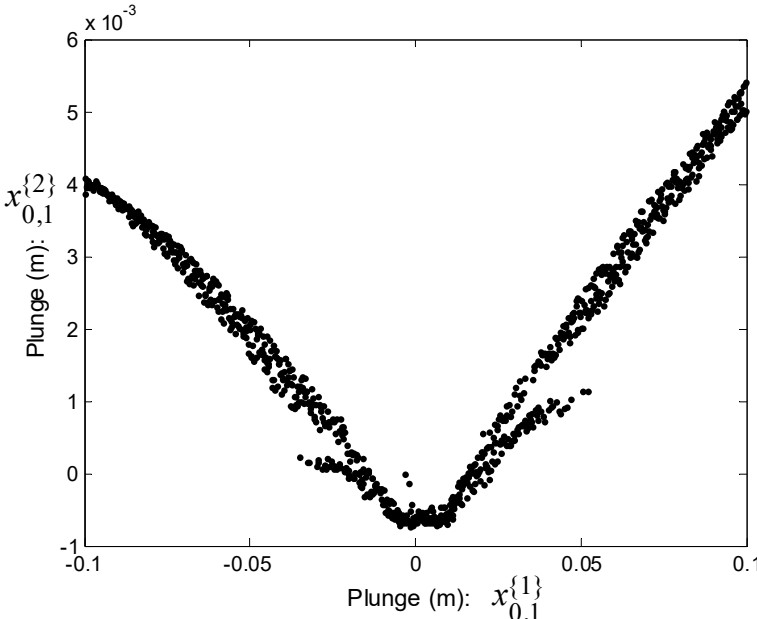

**Figure 18.** Iteration plot for pitch velocity in the first iteration. The airspeed is 20 m/s.

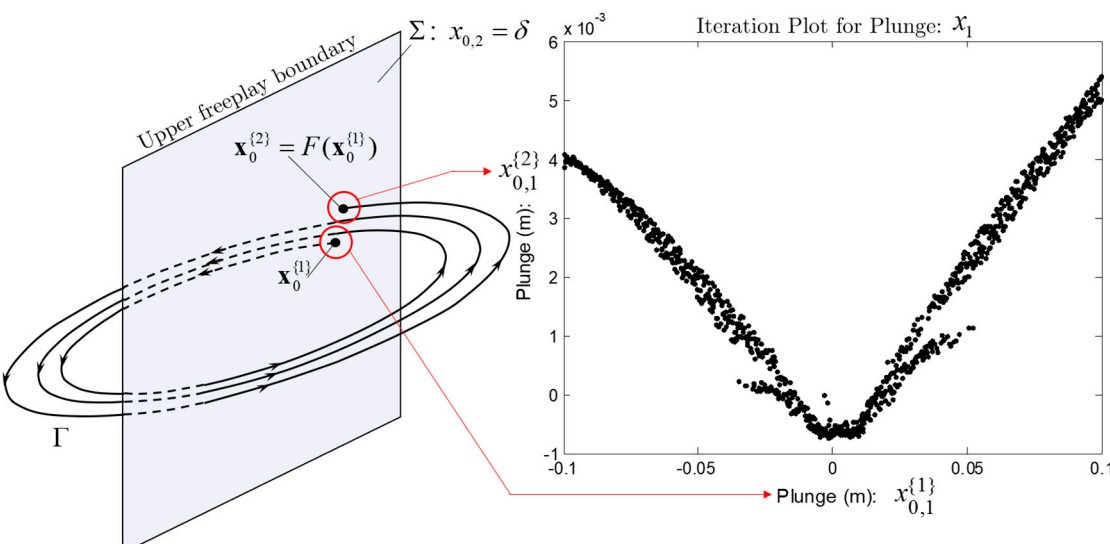

**Figure 19.** An interpretation of Figure 18, connecting the mapping *F* in state space and the iteration plot for pitch velocity. The airspeed is 20 m/s.

Now, we need to check if the initial set $\mathbf{X}_0^{\{1\}}$ specified in Equation (15) is contained in the basins of attraction $\bar{U}_L$. Note that any initial plunge $x_{0,1}^{\{1\}} \in X_{0,1}^{\{1\}} = [-0.1, 0.1]$ will still remain in $X_{0,1}^{\{1\}}$ after the mapping—that is, $x_{0,1}^{\{2\}} \in X_{0,1}^{\{1\}}$. Similar situations are seen in the other system states. Therefore, we confirm that for any $\mathbf{x}_0^{\{1\}} \in \mathbf{X}_0^{\{1\}}$, $\mathbf{x}_0^{\{2\}} = F(\mathbf{x}_0^{\{1\}})$ $\in \mathbf{X}_0^{\{1\}}$, which means that $\mathbf{X}_0^{\{1\}} \subseteq \bar{U}_L$. Thus, we know that there must be some fixed points of $F$ hidden inside region $\mathbf{X}_0^{\{1\}}$.

To determine the fixed points, add a line $l_{\text{equ}}$ that represents $x_{0,1}^{\{1\}} = x_{0,1}^{\{2\}}$ in Figure 18, and find the intersection region $L_0 = R_0 \cap l_{\text{equ}}$, where region $R_0$ consists of $N_S$ points. Then, let $X_{0,1}^{\{2\}}$ be the range of $L_0$ projected on $x_{0,1}^{\{1\}}$, as shown in Figure 20; that is, $X_{0,1}^{\{2\}} = [-7.34 \times 10^{-4}, 5.42 \times 10^{-3}] \subseteq X_{0,1}^{\{1\}}$. A huge contraction is seen in $X_{0,1}^{\{1\}} \to X_{0,1}^{\{2\}}$. Similar procedures are implemented to states $x_{0,3}$, $x_{0,4}$, $x_{0,5}$, and $x_{0,6}$. Finally, we have $\mathbf{X}_0^{\{2\}} \subseteq \mathbf{X}_0^{\{1\}} \subseteq \bar{U}_L$. The procedure of $\mathbf{X}_0^{\{1\}}$ contracting to $\mathbf{X}_0^{\{2\}}$ is called "Iteration-1".

Then, by repeating the procedures above, we can obtain $\mathbf{X}_0^{\{3\}}$ from "Iteration-2", $\mathbf{X}_0^{\{4\}}$ from "Iteration-3", and $\mathbf{X}_0^{\{5\}}$ from "Iteration-4". For the present wing section model, four iterations are sufficient to find the LCO solutions accurately because further implementation of the iterations will show that $\mathbf{X}_0^{\{5\}} \approx \mathbf{X}_0^{\{6\}} \approx \mathbf{X}_0^{\{7\}} \approx \cdots$. Table 1 presents the range of $X_{0,j}^{\{n\}}$ in each "Iteration-$n$" for each system state $x_j$, while Figure 21 illustrates the first four iterations.

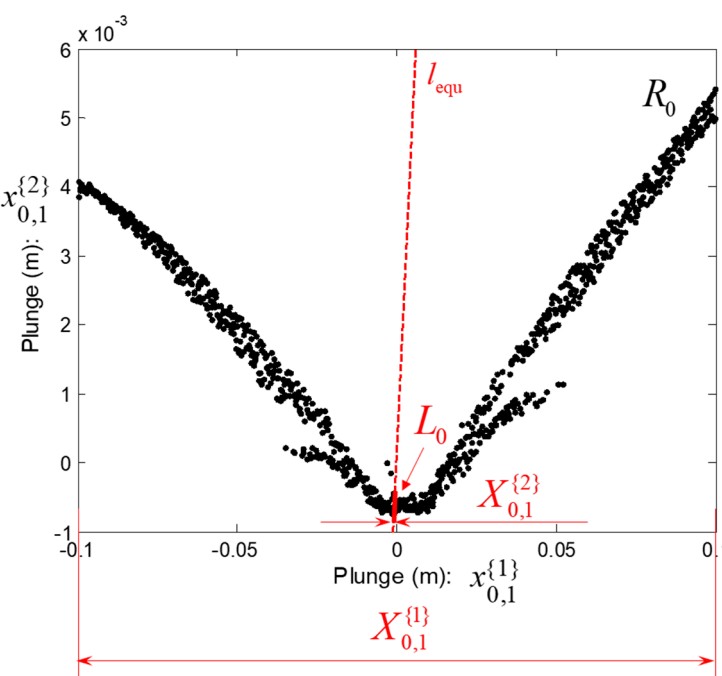

**Figure 20.** Illustration of the range of $X_{0,j}^{\{n\}}$ being contracted to $X_{0,j}^{\{n+1\}}$ by means of the spatial pattern. The airspeed is 20 m/s.

**Table 1.** Ranges of states, relative changes in ranges, and the central processing unit (CPU) time consumed in each iteration. The airspeed is 20 m/s. An iteration will terminate if all ranges vary within 5% compared to those in the last iteration.

| Iteration-$n$ | $x_j$ | | $x_1$, m | $x_3$, m/s | $x_4$, °/s | $x_5$ | $x_6$ | CPU Time |
|---|---|---|---|---|---|---|---|---|
| 1 | $X_{0,j}^{\{1\}}$ | min. | $-0.1$ | $-0.1$ | 0.00 | $-0.1$ | $-0.1$ | — |
| | | max. | $+0.1$ | $+0.1$ | 30.00 | $+0.1$ | $+0.1$ | |
| 2 | $X_{0,j}^{\{2\}}$ | min. | $-7.34 \times 10^{-4}$ | $-6.14 \times 10^{-2}$ | 0.00 | 0.00 | $-3.43 \times 10^{-3}$ | 82.82 s |
| | | max. | $+5.42 \times 10^{-3}$ | $+5.60 \times 10^{-2}$ | 29.98 | $+4.79 \times 10^{-1}$ | 0.00 | |
| | $\Delta\lvert X^{\{1,2\}}\rvert/\lvert X^{\{1\}}\rvert$ | | $-96.92\%$ | $-41.23\%$ | 0.06% | $-8.12\%$ | $-98.29\%$ | |
| 3 | $X_{0,j}^{\{3\}}$ | min. | $-6.83 \times 10^{-4}$ | $-2.49 \times 10^{-2}$ | 0.02 | 0.00 | $-7.44 \times 10^{-4}$ | 78.54 s |
| | | max. | $+5.91 \times 10^{-5}$ | $+1.57 \times 10^{-3}$ | 27.42 | $+2.11 \times 10^{-1}$ | $+1.11 \times 10^{-5}$ | |
| | $\Delta\lvert X^{\{2,3\}}\rvert/\lvert X^{\{2\}}\rvert$ | | $-87.93\%$ | $-77.46\%$ | 8.61% | $-55.81\%$ | $-77.97\%$ | |
| 4 | $X_{0,j}^{\{4\}}$ | min. | $-5.35 \times 10^{-4}$ | $-2.50 \times 10^{-2}$ | 0.00 | 0.00 | $-7.44 \times 10^{-4}$ | 78.66 s |
| | | max. | $+4.97 \times 10^{-5}$ | $+1.48 \times 10^{-3}$ | 12.09 | $+2.12 \times 10^{-1}$ | 0.00 | |
| | $\Delta\lvert X^{\{3,4\}}\rvert/\lvert X^{\{3\}}\rvert$ | | $-21.20\%$ | 0.23% | 55.88% | 0.77% | $-1.39\%$ | |
| 5 | $X_{0,j}^{\{5\}}$ | min. | $-5.04 \times 10^{-4}$ | $-2.52 \times 10^{-2}$ | 0.00 | 0.00 | $-7.47 \times 10^{-4}$ | 78.27 s |
| | | max. | $+6.10 \times 10^{-5}$ | $+1.46 \times 10^{-3}$ | 12.14 | $+2.11 \times 10^{-1}$ | $+1.10 \times 10^{-5}$ | |
| | $\Delta\lvert X^{\{4,5\}}\rvert/\lvert X^{\{4\}}\rvert$ | | $-3.32\%$ | 0.81% | $-0.41\%$ | $-0.30\%$ | 1.86% | |

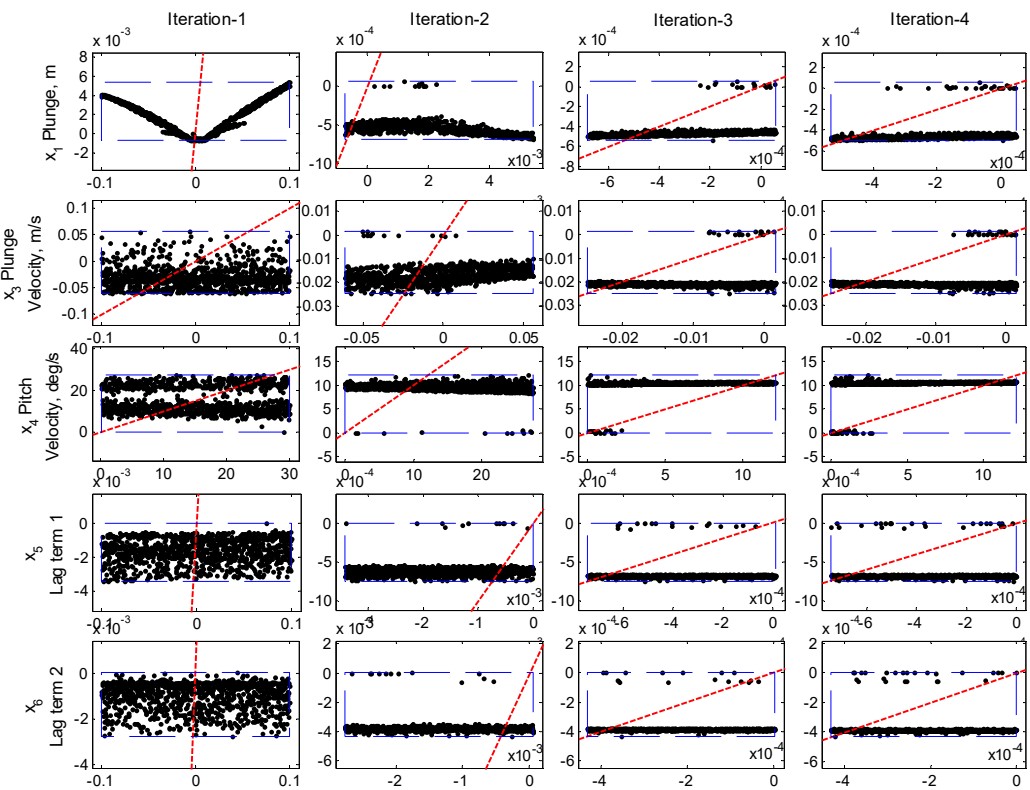

**Figure 21.** Iteration plots for each state in each iterating process, presenting the procedure for the range $X_{0,j}^{\{n\}}$ being contracted to $X_{0,j}^{\{n+1\}}$. Each one of the subplots in this figure is similar to that in Figure 20.

After Iteration-4, clear spatial patterns emerge on each of the five iteration plots, as shown in the rightmost column of Figure 21. For example, we can focus on the iteration plot for $x_{0,4}$, which is located at the third row and fourth column in Figure 21, and re-plot it in Figure 22. Three fixed points can be easily distinguished: one is at $x_{0,4} \approx$ 10.6 deg/s, one is at $x_{0,4} \approx$ 0.45 deg/s, and the last one is at $x_{0,4} =$ 0. The first two fixed points represent two LCO solutions, while the last fixed point indicates a convergence of the system responses. The amplitudes and frequencies are derived from the time integrations of Iteration-4; then, a plot of the LCO amplitude/LCO frequency versus $x_{0,4}$ is given in Figure 23. It is found that the amplitude of the LCO arising at $x_{0,4} \approx$ 10.6 deg/s is larger than 1.0, which indicates that this is a 3-D LCO, and the other LCO is determined as a 2-D LCO, as its relative amplitudes are less than or around 1.0.

As a result, the system under an airspeed of 20 m/s has two LCOs: one is a 3-D LCO that has a non-dimensional amplitude fluctuating within 7.22–7.27 and a frequency of about 4.98 Hz, and the other one is a 2-D LCO whose non-dimensional amplitude is about 0.46 and whose frequency is 4.26 Hz. Moreover, both LCOs are stable since the slopes of the mapping $F$ near the fixed points satisfy $|F'| < 1$, as explained in Section 3.3.

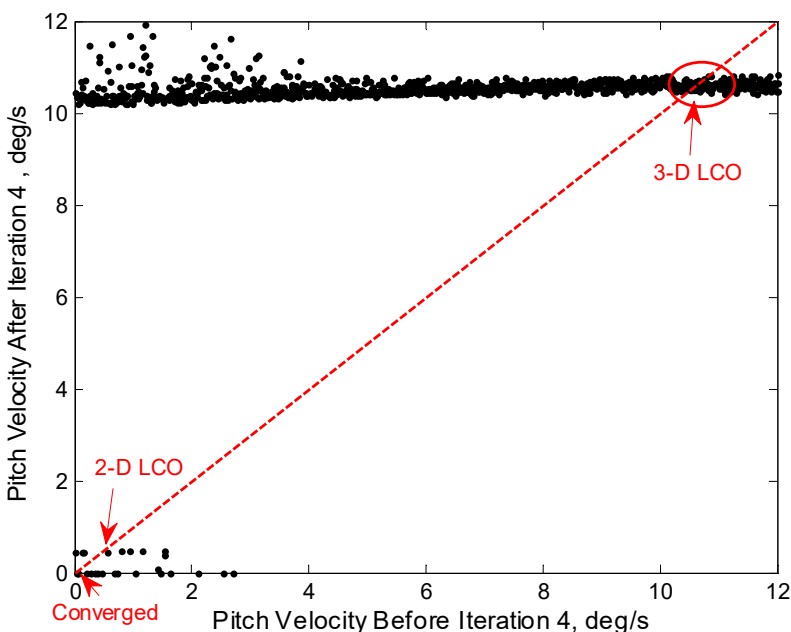

**Figure 22.** The iteration plot of the pitch velocity in Iteration-4, where two kinds of LCOs, as well as the convergence of the system response, are found. The airspeed is 20 m/s.

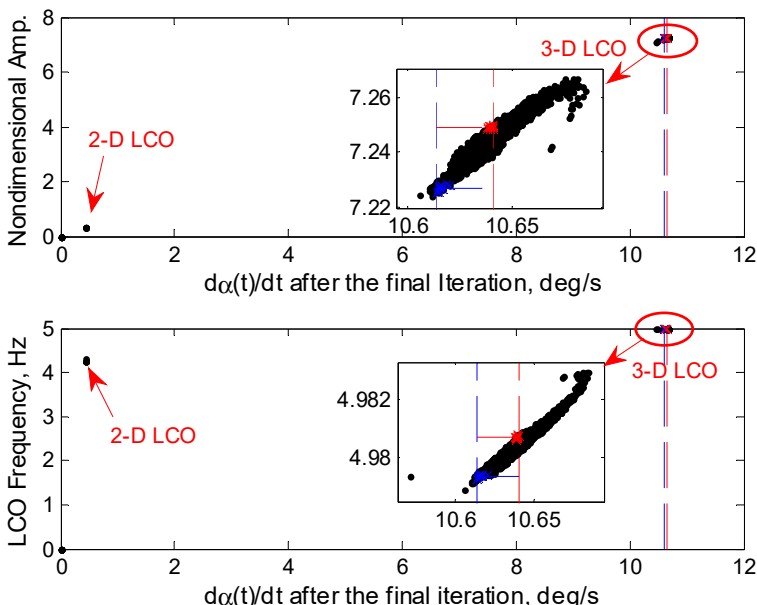

**Figure 23.** LCO amplitudes and dominant frequencies versus pitch velocity for the fourth iteration. The airspeed is 20 m/s.

### 6.2. LCO Results Varying with the Airspeed and a Comparison to Hénon–RK45

Similar procedures are implemented for the 18 cases of airspeeds, which increase from 12.4 to 26.0 m/s. Two branches of LCOs are found, as illustrated in Figures 24 and 25: (1) A 3-D LCO arises at 12.4 m/s, with $\overline{A}_{\mathrm{LCO}}$ and $f_{\mathrm{LCO}}$ fluctuating within 2.48–2.85 and 4.29–4.51 Hz, respectively, depending on the I.C., and ends at about 26.0 m/s, where $\overline{A}_{\mathrm{LCO}}$ surges over 100 and $f_{\mathrm{LCO}}$ increases to 5.22 Hz; (2) a 2-D LCO exists at the airspeeds between 12.2 and 23.0 m/s, where $\overline{A}_{\mathrm{LCO}}$ increases from 0.082 to 0.487 while $f_{\mathrm{LCO}}$ decreases from 4.74 to 3.76 Hz. It is also found that the amplitude and the frequency of the

3-D LCO are quite sensitive to the I.C. at lower airspeeds (about 12.2–13.5 m/s), while such sensitivity almost disappears at higher airspeeds. However, the I.C. almost has no influence on the 2-D LCO. The SSI scheme also confirms that all LCOs are stable with respect to any system state.

The LCO results derived from time integrations with the Hénon–RK45 method and the SSI scheme are compared in Figures 24 and 25. The Hénon–RK45 method uses 7300 I.C.s (see Figure 15), and the LCO results are analyzed based on a simulation time of 15–20 s, which means that the frequency resolution is 0.25 Hz according to Fast Fourier Transformation (FFT), leading to the discontinuity of frequency shown in Figure 25. In the case of airspeed $U > $ 24 m/s, the system response can be a stable LCO, quasi-periodic motion, or chaotic motion, so the points on Figure 24 after $U > $ 24 m/s have a somewhat irregular distribution. However, with respect to the stable 3-D and 2-D LCOs, the results from the SSI scheme agree well with those from the Hénon–RK45 method.

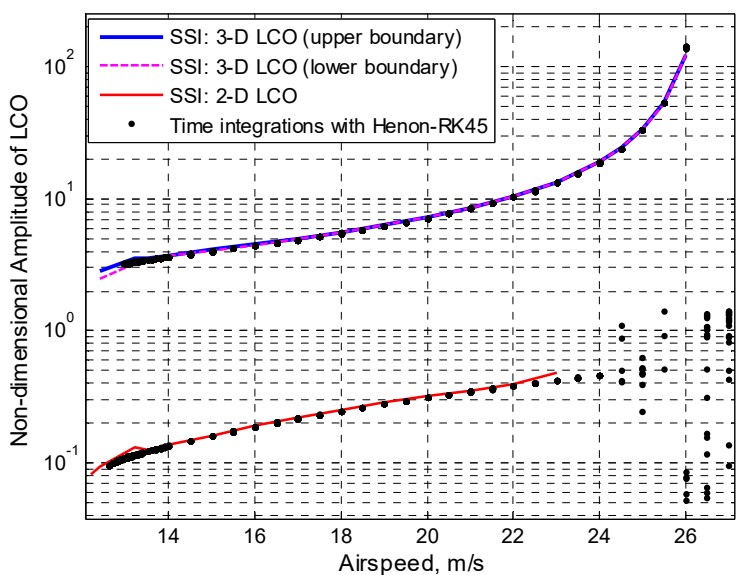

**Figure 24.** Comparison of LCO non-dimensional amplitudes varying with airspeed between the state space iterating (SSI) scheme and the time integrations with the Hénon–RK45 method.

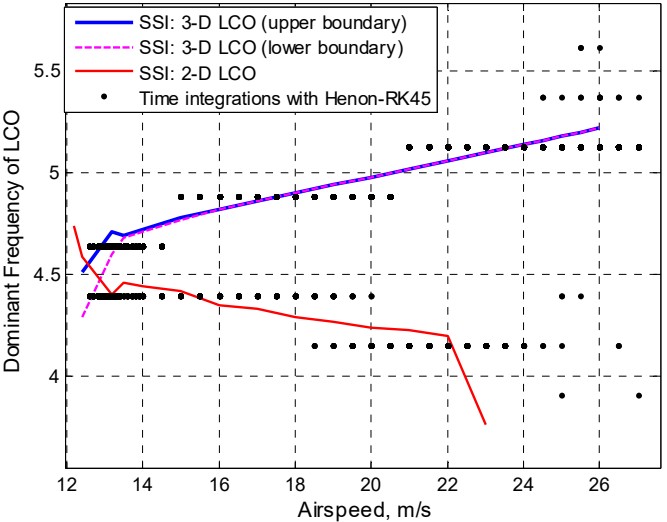

**Figure 25.** Comparison of the LCO dominant frequencies varying with airspeed between the SSI scheme and the time integrations with the Hénon–RK45 method.

## 7. Conclusions

This paper proposed a state space iterating scheme (SSI) to find LCO solutions via a visualization procedure. Inspired by the Poincaré map and the Lorenz map, an $N_T$-time return mapping $F$ was defined and visualized as a set of iteration plots, in which the LCO solutions were represented as the fixed points $\mathbf{x}_L$ of $F$ and were found within a range $\mathbf{X}_0$ that continuously contracts through a series of iterations: $\mathbf{X}_0^{\{1\}} \supseteq \mathbf{X}_0^{\{2\}} \supseteq \cdots \supseteq \mathbf{X}_0^{\{n\}}$. When the range $\mathbf{X}_0^{\{n\}}$ is sufficiently contracted and the spatial patterns shown in the iteration plots are clear enough, the fixed points can be easily distinguished via a visual inspection, and thus, LCOs can be found.

To verify the SSI scheme, a typical plunge–pitch wing section with pitching freeplay was established. Time integrations with a combined fourth–fifth-order Rung–Kutta method and the Hénon method (Hénon–RK45), as well as the proposed SSI scheme, were applied to the model. Both methods determined a three-domain LCO (3-D LCO) and a two-domain LCO (2-D LCO) under a wide range of flow velocities. The SSI scheme confirmed the minimal $\mathbf{X}_0^{\{n\}}$ at the fourth iteration and ultimately found the two LCOs $\mathbf{x}_L \in \mathbf{X}_0^{\{4\}}$ simply through a visual inspection of the iteration plots. The LCO results obtained by SSI were found to be well aligned with those of the Hénon–RK45 method, as shown in Figures 24 and 25.

The highlights of the study are as follows: (1) To obtain the LCO results, Hénon–RK45 checked 100 initial pitch angles for each of the 73 airspeed cases, without guaranteeing that any other arbitrary initial conditions (I.C.s) would lead to the same LCO results, while SSI used 1000 arbitrary I.C.s for each of the 18 airspeed cases, uncovered clear spatial patterns of the system responses, and determined the basins of attraction of the system, which makes the LCO results suitable for any I.C.; (2) the spatial patterns depicted by the SSI scheme could be easily utilized in future studies on other nonlinear responses, such as quasi-periodic motions and chaotic motions, which are usually analyzed via Poincaré sections; (3) the SSI scheme can be easily extended to any other kind of structural nonlinearities and applied to an aeroelastic system with higher dimensions.

However, further studies on SSI are needed, including its application to points (2) and (3) above. We must also determine a better method for selecting arbitrary I.C.s and the parameter $N_T$, which is currently part of our ongoing research.

**Author Contributions:** Conceptualization, data curation, formal analysis, investigation, methodology, software, validation, visualization and writing (original draft) were conducted by X.W. Project administration was conducted by Z.W. Funding acquisition was conducted by C.Y. Writing (review & editing) was conducted by both X.W. and Z.W. Resources and supervision were conducted by both Z.W. and C.Y. All authors have read and agreed to the published version of the manuscript.

**Funding:** Both this research and the APC was funded by the National Key Research and Development Program of China (grant number 2017YFB0503002).

**Institutional Review Board Statement:** Not applicable.

**Informed Consent Statement:** Not applicable.

**Data Availability Statement:** Data is contained within the article or supplementary material.

**Acknowledgments:** This work was supported in part by the National Key Research and Development Program of China under Grant 2017YFB0503002.

**Conflicts of Interest:** The authors declare no conflict of interest.

**Appendix A. Parameters of the Wing Section Model**

This appendix provides the parameters and matrices of the numeric model in Section 3.1.

**Table A1.** Parameters of the wing section.

| Symbol | Explanation | Value | Unit |
|--------|-------------|-------|------|
| $c$ | Chord length | 0.200 | m |
| $b$ | Half chord length | 0.100 | m |
| $L$ | Wing span | 0.400 | m |
| $x_\alpha$ | Distance between flexural axis and central of gravity | 0.010 | m |
| $x_f$ | Distance between leading edge and flexural axis | 0.075 | m |
| $a$ | $a = (x_f - b)/b$ | −0.250 | — |
| $M$ | Gross weight of the wing section | 2.900 | kg |
| $S$ | Static moment: $S = -M x_\alpha$ | −0.029 | kg·m |
| $I_\alpha$ | Inertia moment of the wing section | 0.024 | kg·m² |
| $K_h$ | Linear plunge stiffness | 2372.0 | kg·m/s² |
| $K_\alpha$ | Linear pitch stiffness | 35.50 | kg·m²/s² |
| $C_h$ | Plunge damping | 3.32 | kg·m/s |
| $C_\alpha$ | Pitch damping | 0.04 | kg·m²/s |

The structural modeling procedures are similar to those in Edwards' work [30], which deals with a three-DoF typical wing section:

$$\mathbf{M}_S = \begin{bmatrix} M & S \\ S & I_\alpha \end{bmatrix}, \ \mathbf{D}_S = \begin{bmatrix} C_h & 0 \\ 0 & C_\alpha \end{bmatrix}, \ \mathbf{K}_{S1} = \begin{bmatrix} K_h & 0 \\ 0 & K_\alpha \end{bmatrix}, \ \mathbf{K}_{S0} = \begin{bmatrix} K_h & 0 \\ 0 & 0 \end{bmatrix} \tag{A1}$$

**Table A2.** Matrices in the equations of motion of the wing section model.

| $\mathbf{A}_0$ | | $\mathbf{A}_1$ | | $\mathbf{A}_2$ | |
|--------|--------|--------|--------|--------|--------|
| 0.0081 | 1.2336 | −5.0805 | 1.4583 | −7.4112 | −0.1568 |
| 0.0002 | 0.0308 | −0.1270 | −0.0264 | 0.1289 | −0.0196 |
| $\mathbf{A}_D$ | | $\mathbf{A}_E$ | | $\mathbf{R}$ | |
| 0.9756 | 0.9756 | −0.3265 | −0.2824 | −0.08 | 0 |
| 0.0244 | 0.0244 | −2.3639 | −0.5666 | 0 | −0.60 |

$$\mathbf{A}_{\text{lin}} = \begin{bmatrix} \mathbf{0}_{2\times2} & \mathbf{I}_2 & \mathbf{0}_{2\times2} \\ -\mathbf{M}_A^{-1}\mathbf{K}_{A0} & -\mathbf{M}_A^{-1}\mathbf{D}_A & \tfrac{1}{2}\rho U^2 \mathbf{M}_A^{-1}\mathbf{A}_D \\ \mathbf{0}_{2\times2} & \mathbf{A}_E & U/b \cdot \mathbf{R} \end{bmatrix} \tag{A2}$$

$$\mathbf{A}_{NL} = \begin{bmatrix} \mathbf{0}_{2\times 2} & \mathbf{I}_2 & \mathbf{0}_{2\times 2} \\ -\mathbf{M}_A^{-1}\mathbf{K}_{A1} & -\mathbf{M}_A^{-1}\mathbf{D}_A & \frac{1}{2}\rho U^2 \mathbf{M}_A^{-1}\mathbf{A}_D \\ \mathbf{0}_{2\times 2} & \mathbf{A}_E & U/b \cdot \mathbf{R} \end{bmatrix} \tag{A3}$$

where

$$\begin{cases} \mathbf{M}_A = \mathbf{M}_S - \frac{1}{2}\rho b^2 \mathbf{A}_2 \\ \mathbf{D}_A = \mathbf{D}_S - \frac{1}{2}\rho b U \mathbf{A}_1 \\ \mathbf{K}_{A0} = \mathbf{K}_{S0} - \frac{1}{2}\rho U^2 \mathbf{A}_0 \\ \mathbf{K}_{A1} = \mathbf{K}_{S1} - \frac{1}{2}\rho U^2 \mathbf{A}_0 \end{cases} \tag{A4}$$

and

$$\mathbf{b} = \begin{bmatrix} \mathbf{0}_{2\times 1} \\ \mathbf{M}_A^{-1}\begin{bmatrix} 0 \\ 1 \end{bmatrix} \\ \mathbf{0}_{2\times 1} \end{bmatrix} \tag{A5}$$

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
