# Peer review of "Integration of Freeplay-Induced Limit Cycles Based On a State Space Iterating Scheme"

_applsci, doi:10.3390/app11020741_

Round 1

Reviewer 1 Report

The topic of the paper is a novel method of deailing with freeplay in aeroelastic system. As such, the topic is well aligned with the scope of the journal. My specific comments are the following:

  • Can the proposed method be extended to deal with other kinds of nonlinearities? For example the nonlinear 6 DoF equations of motion?
  • The quality of the plots should be increased for better readability.
  • Are two lag states enough for the RFA. Typically, aeroelastic systems contain many lag states which makes control design difficult.
  • Is there a come up with a more structured set of primary I.C. S_ini and not just randomly select N_s points? This could help reduce computational time.
  • How can it be ensured that no "worst-case" I.C. is skipped in the analysis?

I recommend revising the paper based on these recommendation.

Reviewer 2 Report

Reviewers' comments:

Manuscript number: applsci-1037691

Title: Integration of Freeplay-Induced Limit-Cycle Based on A State-space Iterating Scheme.

Comments: 

The manuscript reported on Integration of Freeplay-Induced Limit-Cycle Based on A State-space Iterating Scheme. The manuscript needs a detailed editing. It cannot be recommended for publication in the present form. I hope the following points would be helpful for the authors.

- Language needs substantial improvement. Please consult a native English speaker or a language editing service.

- Qualitative information’s are missing in abstract.

- The introduction section should be improved; more related papers must be discussed and superiority, novelty, critical improvement in this study must be clarified.

- Please provides the references for all equations and formula.

- Figure 1, not clear make clear.

- Figure 3, not clear make clear.

- Figure 8, not clear make clear.

- Figure 12, Iteration plots for each state in each iterating process. Not clear make clear.

- Several faults: are added or missing spaces between words: see manuscript file.

- The conclusion part should rebuild to let it fluent.

- References: there are recent references in 2019 and 2020 treating the same subject, you can use and make all references in same format for volume number, page numbers and journal name, because it is difficult to searching and reading.

Based on these, I advise the authors to rectify the above-mentioned errors and we hope to re-evaluate the revised manuscript.

Round 2

Reviewer 1 Report

The authors addressed all my comments of the previous version in a detailed answer. I recommend accepting the paper once the English of the paper is improved in the way the authors wrote they would do it. There are no further technical details that need improvement from my side.

Reviewer 2 Report

Reviewers' comments:

The authors revised the manuscript according to the reviewers' comments.

So that I recommended this manuscript accept for publication in Applied Sciences.
